# The Low-Rank Simplicity Bias in Deep Networks

**Minyoung Huh**                                                            *minhuh@mit.edu*
*MIT CSAIL*

**Hossein Mobahi**                                                          *hmobahi@gmail.com*
*Google Research*

**Richard Zhang**                                                          *rizhang@adobe.com*
*Adobe Research*

**Brian Cheung**                                                            *cheungb@mit.edu*
*MIT CSAIL & BCS*

**Pulkit Agrawal**                                                          *pulkitag@mit.edu*
*MIT CSAIL*

**Phillip Isola**                                                            *phillipi@mit.edu*
*MIT CSAIL*

**Reviewed on OpenReview:** *https://openreview.net/forum?id=bCiNWDmlY2*

## Abstract

Modern deep neural networks are highly over-parameterized compared to the data on which they are trained, yet they often generalize remarkably well. A flurry of recent work has asked: why do deep networks not overfit to their training data? In this work, we make a series of empirical observations that investigate and extend the hypothesis that deeper networks are inductively biased to find solutions with lower effective rank embeddings. We conjecture that this bias exists because the volume of functions that maps to low effective rank embedding increases with depth. We show empirically that our claim holds true on finite width linear and non-linear models on practical learning paradigms and show that on natural data, these are often the solutions that generalize well. We then show that the simplicity bias exists at both initialization and after training and is resilient to hyper-parameters and learning methods. We further demonstrate how linear over-parameterization of deep non-linear models can be used to induce low-rank bias, improving generalization performance on CIFAR and ImageNet without changing the modeling capacity.

## 1 Introduction

It has become conventional wisdom that the more layers one adds, the better a deep neural network (DNN) performs. This guideline is supported, in part, by theoretical results showing that deeper networks can require far fewer parameters than shallower networks to obtain the same modeling "capacity" (Eldan & Shamir, 2016). While it is not surprising that deeper networks are more expressive than shallower networks, the fact that state-of-the-art deep networks do not overfit, despite being heavily over-parameterized, defies classical statistical theory (Geman et al., 1992; Zhang et al., 2017; Belkin et al., 2019) – e.g., Dosovitskiy et al. (2020) trains a 632 million parameter, 200+ layer model, on 1.3 million images.

The belief that *over-parameterization via depth improves generalization* is used axiomatically in the design of neural networks. Unlike conventional regularization methods that penalize model complexity (e.g., $\ell_1/\ell_2$ penalty), over-parameterization does not. Yet, like explicit regularization, over-parameterization appears to prevent the model from over-fitting (Belkin et al., 2018; Nakkiran et al., 2019a). While there has been

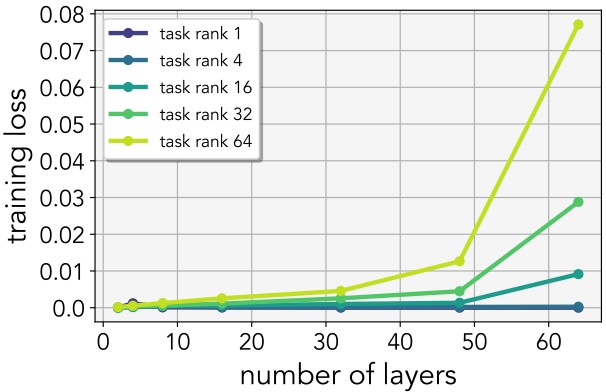

Figure 1: **Deep nets *struggle* to fit high-rank *linear* functions:** We report the training loss of neural networks of different depths optimized to solve linear regression. The rank of the underlying linear function is varied in the range $[1, 64]$. While shallow networks achieve zero training loss, the training loss worsens with increased depth and task rank (see Appendix F for training details).

an extensive effort to analyze the effect of the *implicit regularization* of over-parameterization on neural networks (see Section 6), prior investigations have been mostly limited to linear models for theoretical analysis or have been left as an under-explored side observation. This work aims to further the existing efforts by providing extensive empirical experiments and analysis on linear and non-linear networks for practical learning paradigms.

Our analysis begins with a non-intuitive observation that *over-parameterization* hurts the ability to overfit simple linear functions. We trained ReLU networks with varying depths on a set of linear regression tasks $Y = W^*X$. For some randomly sampled $X$, we minimize the least-squares error between the prediction $\hat{Y}$ and the ground-truth targets $Y$. In Figure 1, we plot the converged loss when varying the depth of the model and the underlying rank of the task: $\text{rank}(W^*) = \{1, 4, 16, 32, 64\}$. The results reveal that deeper networks touted for their ability to model complex functions struggle to fit even (high-rank) linear functions. In contrast, shallower networks perfectly minimize the loss.

One explanation of these results is improper optimization of neural network parameters. We used standard SGD based optimizers and experimented with a wide range of hyper-parameters that we detail in Appendix F. While there may exist an optimization algorithm that can perfectly minimize training error, we do not know of such an optimizer. At first, our result might seem to be at odds with the work of Zhang et al. (2017) observing that deep networks (8 layers) can achieve zero training error on random data. However, our results are consistent because Zhang et al. (2017) did not experiment with deeper networks, and predicting labels from images is (loosely speaking) not a full rank prediction problem.

The second possibility is our hypothesis that *deep over-parameterized networks are biased to find low effective rank solutions.* Results in Figure 1 corroborate this hypothesis, but the problem is that the concept of rank is not defined for a non-linear network. However, it is still possible to study the effective rank of the feature embeddings learned by the penultimate layer of the neural network. In the case of a linear neural network, the embedding and parameter rank are equivalent. In the remainder of this work, we probe the relationship between the effective rank of the embedding and depth. Our findings indeed strengthen the hypothesis that deeper networks find lower effective rank solutions.

Prior work has shown that over-parameterized linear networks find minimum norm solutions (Gunasekar et al., 2017; Arora et al., 2019a), which in special cases, is equivalent to finding low-rank solutions. Valle-Perez et al. (2019) also suggested that deep *non-linear* networks are "simple functions", but does not make any connection to the depth of the network nor explain why the model would likely converge to a "simple function". Here, "simple function" is measured by the Lempel-Ziv complexity of the output from a randomly initialized boolean network. Our work ties together these two lines of research by investigating how the hypothesis space of the network changes when the network is over-parameterized with depth. We specifically study the relationship between *the rank of the embedding* – the effective rank computed on the linear kernel of the network's output features – and depth for both *linear* and *non-linear* networks.

The fact that deeper networks are primed to learn solutions that have low effective rank embedding may also explain why they generalize despite being over-parameterized – most natural data (e.g., images) actually lies

on a low-dimensional manifold, and common problems such as classification require predicting quantities that are much lower-dimensional than the inputs.

In summary, this work provides a new set of observations that expand the growing body of work on over-parameterization. Mainly, we make a series of empirical observations that indicate deep nets have an inductive bias to find lower rank embeddings.

- We observe that deep nets, *even at initialization*, are biased to map data into low-rank embeddings. We observed this bias to exist after training with gradient descent.
- We observe that the bias towards low-rank embeddings exists in a wide variety of common optimizers, *even* those that do not use gradient descent (e.g., random-search).
- We find that even if we initialize the networks to be low or high rank, the effective rank of the converged solution is largely dependent on the depth of the model.
- This set of observations leads us to conjecture that *deeper networks are implicitly biased to find lower effective rank embeddings because the volume of functions that map to low effective rank embeddings increases with depth.*
- We leverage our observations to demonstrate linear over-parameterization by "depth" can be used to achieve better generalization performance on CIFAR (Krizhevsky et al., 2009) and ImageNet (Russakovsky et al., 2015) *without* increasing modeling capacity.

## 2 Preliminaries

### 2.1 Neural networks and Over-parameterization

**Simple linear network**
A simple linear neural network transforms input $x \in \mathbb{R}^{n \times 1}$ to output $\hat{y} \in \mathbb{R}^{m \times 1}$, with a learnable parameter matrix $W \in \mathbb{R}^{m \times n}$,

$$\hat{y} = Wx. \tag{1}$$

For notational convenience, we omit the bias term.

**Over-parameterized linear networks**
One can over-parameterize a *linear* neural network by defining $d$ matrices $\{W_i\}_{i=1}^d$ and multiplying them successively with input $x$:

$$\hat{y} = W_d W_{d-1} \cdots W_1 x = W_e x, \tag{2}$$

where $W_e = \prod_{i=1}^d W_i$. As long as the matrices are of the correct dimensionality — matrix $W_d$ has $m$ columns, $W_1$ has $n$ rows, and all intermediate dimensions $\{\dim(W_i)\}_{i=2}^{d-1} \geq \min(m, n)$ — then this over-parameterization expresses the same set of functions as a single-layer network. We disambiguate between the collapsed and expanded set of weights by referring to $\{W_i\}$ as the over-parameterized weights and $W_e$ as the *end-to-end* or the *effective weights*.

**Non-linear networks**
For *non-linear* network, activation function $\psi$ (e.g. ReLU) is interleaved between the weights:

$$\hat{y} = W_d \psi(W_{d-1} \ldots \psi(W_1(x))) \tag{3}$$

In contrast to linear networks, non-linear models become more expressive as more layers are added.

### 2.2 Effective rank

We characterize the rank of a matrix using a continuous measure known as the *effective rank*:

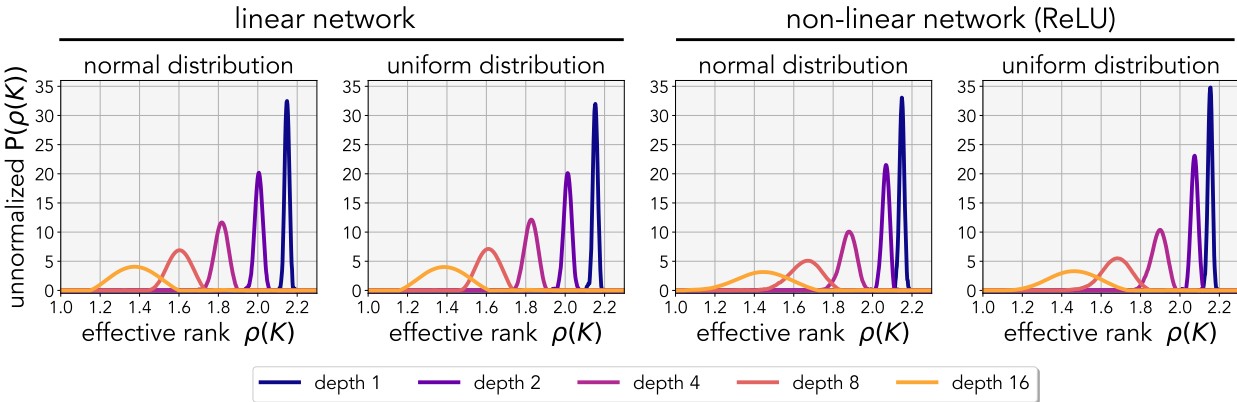

Figure 2: **Deep networks are biased toward low effective rank:** The approximated probability density function (PDF) of the effective rank $\rho$ over the Gram matrix is computed from features of the networks. The Gram matrix is computed with 256 random inputs, and we use 4096 network parameter samples to approximate the cumulative distribution function. The CDF is used to compute the PDF via the finite difference method. We apply Savitzky & Golay (1964) filter to smoothen out the approximation. There exists more probability mass for lower effective rank embeddings when adding more layers. The experiment is repeated for both normal and uniform distributions. For linear networks, the effective parameters are fixed across depth, while for non-linear networks, this is not the case.

**Definition 2.1** (Effective rank). *(Roy & Vetterli, 2007) For any matrix $A \in \mathbb{R}^{m \times n}$, the effective rank $\rho$ is defined as the Shannon entropy of the normalized singular values:*

$$\rho(A) = - \sum_{i=1}^{\min(n,m)} \bar{\sigma}_i \log(\bar{\sigma}_i),$$

*where $\bar{\sigma}_i = \sigma_i / \sum_j \sigma_j$ are normalized singular values, such that $\sum_i \bar{\sigma}_i = 1$. Also referred to as the spectral entropy. Without loss of generality, we drop the exponentiation for convenience.*

This measure gives us a meaningful representation of "continuous rank", which is maximized when the magnitude of the singular values are all equal and minimized when a single singular value dominates relative to others. The effective rank provides us with a metric that summarizes the distribution envelope. Effective rank has been used in prior works (Arora et al., 2019a; Razin & Cohen, 2020; Baratin et al., 2021) and we use this measure extensively throughout our work. We have also found that our observations are consistent with the closest definition of rank in which we threshold the smallest singular values after normalization (Appendix D).

## 2.3 Embedding maps

A parameteric function $f_{\{W\}} \in \mathcal{F}_{\mathcal{W}}$ is a neural network parameterized with a set weights $\{W\} = \{W_1, \ldots, W_d\}$ that maps the input space to the output space $\mathcal{X} \to \mathcal{Y}$. For a dataset of size $q$, the input and output data is $X \in \mathbb{R}^{n \times q}$ and $Y \in \mathbb{R}^{m \times q}$. Then, the predicted output is $\hat{Y} = W_d \psi(\Phi) = f_{\{W\}}(X)$, where $\Phi \in \mathbb{R}^{n' \times q}$ is the last-layer embedding and $W_d \in \mathbb{R}^{m \times n'}$ is the last layer of the network.

We analyze the embedding space by computing the effective rank on the Gram/kernel matrix $K \in \mathbb{R}^{p \times p}$ where $p$ is the size of the test set. The $ij$-th entry of the Gram matrix corresponds to a distance kernel $K_{ij} = \kappa(\phi_i, \phi_j)$ where $\phi_i$ corresponds to the $i$-th column of $\Phi$. We use the model's intermediate features before the linear classifier and use cosine distance kernel: $\kappa(\phi_i, \phi_j) = \frac{\phi_i \phi_j^T}{\|\phi_i\| \|\phi_j\|}$, a common method for measuring distances in feature space (Kiros et al., 2015; Zhang et al., 2018). We observed our findings to be consistent with other common choices of dot-product distance functions such as linear kernels and correlation kernels (Appendix D).

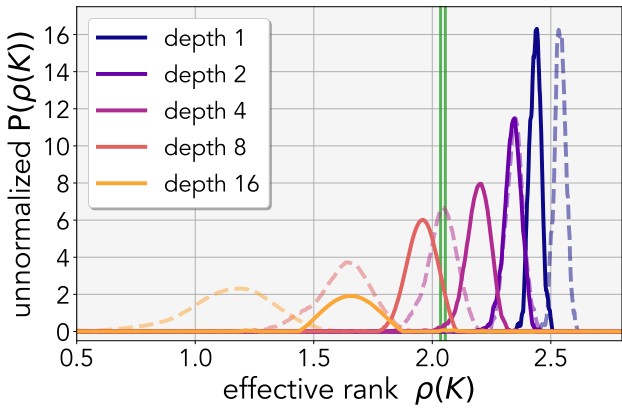

Figure 3: **Distribution of non-linear nets at convergence**: Rank distribution after training the network to zero-training error with gradient descent. The dotted line indicates the initial distribution, the solid line indicates the converged distribution, and the green line indicates the task rank. Despite all models having the same functional capacity, the model's ability to find the underlying solution depends on the original parameterization of the network. Despite all models achieving zero-training error, models of different depth recover different underlying solutions. In this experiment, the model with a depth of 4 or 8 finds a better generalizing solution on a held-out set than models with more or fewer layers.

The dimensionality of the Gram-matrix depends on the data samples and does not depend on the model parameters. For non-linear networks, we make comparisons at the zero training error regime.

Gram matrices are often used to analyze the optimization and generalization properties of neural networks (Zhang et al., 2019; Du et al., 2018; 2019; Wu et al., 2019; Arora et al., 2019b). In natural data, it is often assumed that we are trying to discover a low-rank relationship between the input and the label. For example, a model that overfits every training sample without inferring any structure on the data will generally have a test gram-matrix that is a higher rank than that of a model that has learned parsimonious representations. A lower rank on held-out data indicates less excess variability and is indicative of studying generalization and robustness. The intuition becomes clearer in linear networks since the rank of the Gram matrix depends on the rank of the linear transformation computed by the network. We illustrate this empirically in Appendix L, where we see that there is a tight relationship between the effective-rank of the linear weight matrix and the effective-rank of the resulting Gram matrix.

## 2.4 Least squares

Given a dataset $X, Y$ generated from $W^*$, the goal is to regress a parameterized function $f_{\{W\}}(\cdot)$ to minimize the squared-distance $\|f_{\{W\}}(X) - Y\|_2^2$. The rank$(W^*)$ is a measure of the "intrinsic dimensionality" of the data, and we refer to it as the *task rank*. In this work, we exclusively operate in the under-determined regime where we have fewer training examples than model parameters. This ensures that there is more than one minimizing solution.

## 3 The parameterization bias of depth

Given that our models can always fit the data, what are the implications of searching for the solution in the over-parameterized model? In linear models, this is equivalent to searching for solutions in $\{W_i\}$ versus directly in $W_e$. One difference is that the gradient direction $\nabla_{\{W_i\}}\mathcal{L}(\{W_i\})$ is usually different than $\nabla_{W_e}\mathcal{L}(W_e)$ for a typical loss function $\mathcal{L}$ (see Appendix J). The consequences of this difference have been previously studied in linear models by Arora et al. (2018; 2019a), where the over-parameterized update rule has been shown to accelerate training and encourage singular values to decay faster, resulting in a low nuclear-norm solution. Here we motivate a result from the perspective of parameter volume space.

**Conjecture 3.1.** *Deeper networks have a greater proportion of parameter space that maps the input data to lower-rank embeddings; hence, deeper models are more likely to converge to functions that learn simpler embeddings.*

We now provide a set of empirical observations that supports our conjecture. Our work and existing theoretical works on gradient descent biases are *not* mutually exclusive and are a likely complement. We emphasize that

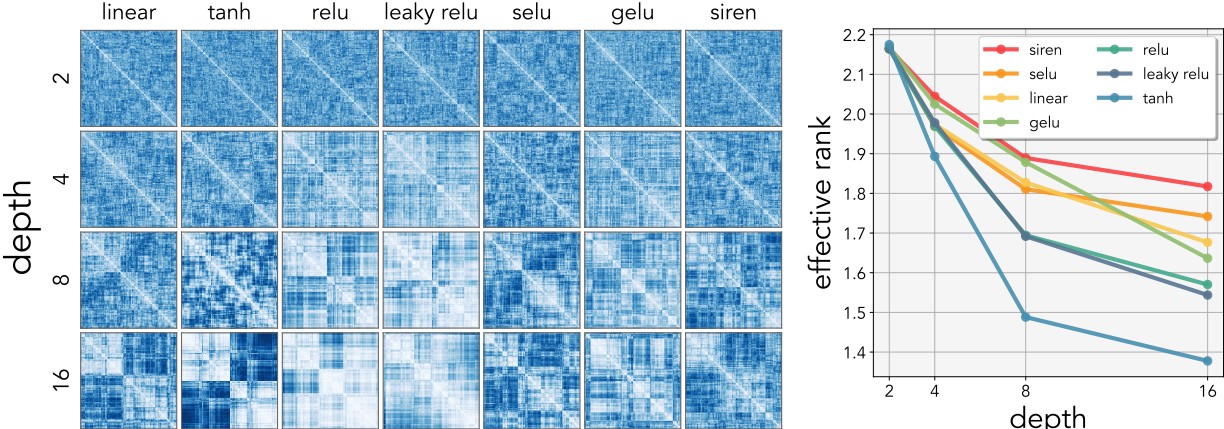

Figure 4: **Gram matrices of networks:** Gram matrices of neural networks trained with various non-linearities and depth. Since increasing the number of non-linear layers increases the functional expressivity of the network, the Gram matrix is computed using the cosine distance on the features of the test set near zero-training loss. Increasing the number of layers decreases the effective rank of the Gram matrix on a variety of non-linear activation functions. The Gram matrix is hierarchically clustered (Rokach & Maimon (2005)) for visualization. We observe the emergence of block structures in the Gram matrix as we increase the number of layers, indicating that the embeddings become lower rank with depth.

we do *not* make any claims on the simplicity of the function, but only on the simplicity – lower effective rank – of the embeddings.

## 3.1 Low-rank simplicity bias of deep networks

**Observation 3.1.** *Randomly initialized deep nets are biased to correspond to Gram matrices with a low effective rank.*

When sampling random neural networks, both linear and non-linear, we observed that the Gram matrices computed from deeper networks have a lower effective rank. We quantify this observation by computing the distribution over the effective rank of the Gram matrix in Figure 2. Here, the weights of the neural networks are initialized using uniform $W_i \sim \mathcal{U}(\cdot, \cdot)$ or Normal distributions $W_i \sim \mathcal{N}(\cdot, \cdot)$. The input, output, and intermediate dimensions are 32, giving parameters $\{W_i\} \in \mathbb{R}^{d \times 32 \times 32}$ for a network with $d$ layers. We draw 4096 random parameter samples and compute the effective rank on the resulting Gram matrix. We see that the distribution density shifts towards the left (lower effective rank) when increasing the number of layers. These distributions have a small overlap and smoothen out with increased depth. This observation shows that depth correlates with lower effective rank embeddings.

The low-rank bias becomes more intuitive in linear models as there is a simple way to relate the Gram matrix to the weights of the model $K \approx (W_{d-1:1}X)^T(W_{d-1:1}X)$. Intuitively, if any constituent matrices are low-rank, then the product of matrices will also be low-rank – the product of matrices can only decrease the rank of the resulting matrix: $\mathrm{rank}(AB) \leq \min(\mathrm{rank}(A), \mathrm{rank}(B))$ (Friedberg et al., 2003). In Appendix L, we show that as the depth of the model increases, both the effective rank of the Gram matrix and the weights decrease together. Another way to interpret our observation is that for linear models, over-parameterization does not increase the expressivity of the function but re-weights the likelihood of a subset of parameters – the hypothesis class. For non-linear models, we *cannot* make the same claims.

Although uniform sampling under the parameter distribution is an unbiased estimator of the volume of the parameter space, it is certainly possible that a sub-space of the parameters is more likely to be observed under gradient descent. Hence, by naively sampling networks, we may never encounter model parameters

that gradient descent explores. In light of this, we repeat our experiment above by computing the PDF on randomly sampled parameters after taking $n$ gradient descent steps.

**Observation 3.2.** *Deep neural networks trained with gradient descent also learn to map data to simple embedding with low effective rank.*

Figure 3 illustrates the change in distribution as we train our model to convergence using gradient descent. Each randomly drawn network sample is trained to minimize the least-squares error. The initial distribution is plotted with dotted lines, and the converged distribution is plotted with solid lines. As the model is trained, the distribution of the rank shifts towards the ground-truth rank (green line) but is constrained by the depth of the model. We highlight that while the observation would have been trivial and expected if the model recovered the exact ground-truth rank at zero-training error. However, the surprising observation is that even if these models achieved zero-training error, the effective rank of the recovered solution depends on the depth of the network – deeper models find lower effective rank solutions, implying that generalization properties would vary based on the parameterization of the models. Since the observed bias stems from the model's parameterization, the same bias must also exist under other common and natural choices of optimizers. We investigate this claim in the next section.

In Figure 4, we further visualize the learned Gram matrices when varying the depth of the model. The Gram matrices trained with various non-linear activation functions also emit the same low-rank simplicity bias. These activation functions include standard functions such as ReLU and Tanh as well as recently popularized non-linear functions such as GeLU ((Hendrycks & Gimpel, 2016)), and the sinusoidal activation function from SIREN ((Sitzmann et al., 2020)). By hierarchically clustering (Rokach & Maimon, 2005) these Kernels, we can directly observe the emergence of block structures in the Gram matrices as we increase the number of layers, implying that the embeddings become lower rank with depth.

### 3.2 Is the low-rank bias specific to gradient descent?

**Observation 3.3.** *Deep neural networks trained with common and natural choices of optimizers also exhibit the low-rank embedding bias.*

The low-rank bias of deep networks has been primarily studied under the context of first-order gradient decent (Arora et al., 2018; 2019a): *how and why does gradient descent converge to low nuclear norm solution.* In contrast, our conjecture focuses on the bias of parameterization of the network and *not* on the bias introduced by the gradient descent. Since parameterization bias exists regardless of the optimizer choice, we would expect to observe the low-rank simplicity bias on a wide range of optimizers. We directly show this in Figure 5 by ablating across various popular choices of optimizers on least-squares with linear networks. Here, we compare against Nesterov (Nesterov (1983); momentum), ADAM (Kingma & Ba (2015); hessian approximator), L-BFGS (Liu & Nocedal (1989); second-order), CMA-ES (Hansen et al. (2003); evolutionary-search), and random search. All models were trained to zero training error except for random search. For random search, we randomly initialize the network $100,000$ times and take the best performing sample. As we have seen with gradient descent, the experiment indicates that even when we train with a wide suite of commonly used optimizers, the solution obtained by these models depends on how the model was originally parameterized.

### 3.3 Can the bias be explained solely by initialization?

The previous set of experiments indicates that deeper networks are biased towards low effective-rank embedding at both initialization and convergence. In these experiments, the random settings of neural networks had different initial distributions. This happens because, even if the individual weights are normally distributed, the weights constructed from a series of matrix multiplications result in a distribution that has a high density around zero. For example, the product of 2 normally distributed weights becomes symmetric $\chi$-squared distribution, with 1 degrees of freedom. Hence, one could argue that the converged solutions have low effective rank because of the initialization bias.

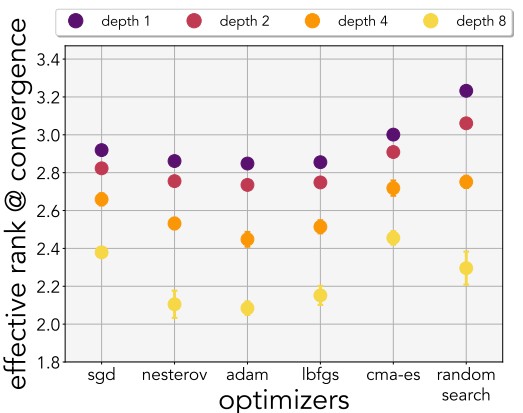

Figure 5: **Low-rank bias & optimizers**: Least-squares trained on linear neural networks using various optimization methods. The rank of the converged Gram matrix is correlated with the depth of the network. The experiment is repeated 5 times. Except for Random Search, all models achieve 0 training loss. While the solution achieved depends on the optimizer, the underlying low-rank bias of depth persists across optimizers and is not specific to gradient descent. All models have the same functional expressivity.

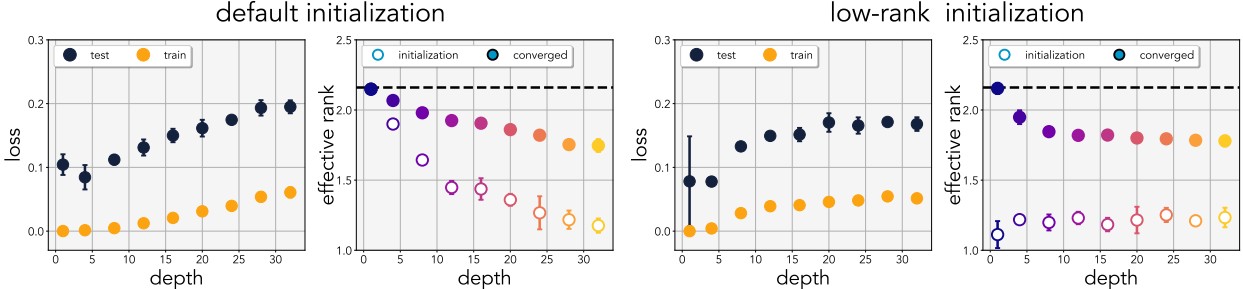

Figure 6: **Bias of parameterization:** The effective rank of the Gram matrix from initialization to convergence on various depth. For each depth, we train a linear network using gradient descent on least squares regression. We repeat our experiments 5 times with different seeds, and we report the median of these runs. The rank at initialization and convergence is indicated by white and colored dots, respectively. For deeper models, the effective-rank is lower at initialization because the product of normally distributed weights is no longer normally distributed. On the right, we initialize the networks with the same low-rank distribution of weights as the 32-layer model. We observe that shallower networks tend to converge to higher rank embeddings. All models in this experiment have the same functional expressivity. While it may seem that non-zero training error at high-depth is under-fitting due to poor optimization choices, we exhaustively search over the optimization hyper-parameters. We list the optimization choices in the A.

**Observation 3.4.** *Deep neural networks are biased towards learning low effective-rank embeddings and are insensitive to initialization.*

To test whether the initialization of the model affects the effective rank of the converged solution. We optimize our network $W \in \mathbb{R}^{d \times 32 \times 32}$ on least-squares where the task-rank is set to 24. All models are trained for 4000 epochs using gradient descent, and the best learning rate is chosen for each depth. In Figure 6 (left), for models using *default initialization*, we show that increasing the number of layers decreases the effective rank of the Gram matrix at convergence. We repeat the experiment in Figure 6 (right) by initializing the over-parameterized models with the distribution associated with the 32-layer linear network. Following a similar trend to that of default initialization, we observe that deeper models learn embeddings that are a lower effective rank than the shallower counterparts. Although initialization is not insignificant, we see that the depth of the model has tight control over the solution which the model explores. For deeper networks, the majority of the parameter volume is mapped to low effective rank embedding (Observation 3.1), and therefore it is expected that a typical search algorithm would likely encounter parameters that map to low effective rank embeddings regardless of initialization. Similarly, for a shallower network, it would be easier to find a solution with higher effective rank embeddings.

### 3.4 Relation to random matrix theory

In linear models, we have a special case in which the low-rank embedding corresponds to low-rank weights. This enables us to make a natural connection to existing theoretical work from random matrix theory (RMT), which studies the spectral distribution under matrix multiplications (Akemann et al., 2013a;b; Burda et al., 2010). We leverage the results from (Pennington et al., 2017; Neuschel, 2014) to show the following:

**Theorem 3.1.** *Let $\rho$ be the effective rank measure defined in Definition 2.1. For a linear neural network with d-layers, where the parameters are drawn from the same Normal distribution $\{W_i\}_{i=1}^d \sim \mathcal{W}$, the effective rank of the weights monotonically decreases when increasing the number of layers when $dim(W) \to \infty$,*

$$\rho\left(W_d W_{d-1} \ldots W_1\right) \le \rho\left(W_{d-1} \ldots W_1\right)$$

*Proof.* See Appendix H.

Given the current set of mathematical tools, our preliminary theory depends on many assumptions, such as infinite width networks and the distribution of the weights; this is akin to many existing theoretical works. Yet, we have observed in practice that the empirical spectral distribution of finite-width models is well approximated by random matrix theory (see Appendix B) in practice. We emphasize that the main contribution of our work is on the empirical theory of the low-rank bias of deep networks; nonetheless, we show that there is a natural theoretical connection to RMT in hopes of stimulating future works.

## 4 Over-parameterization as a regularizer

Thus far, we have observed that depth acts as a bias for finding functions with low effective rank embeddings. As one could imagine, this inductive bias of depth could be used to help but also hurt generalization performance. Our observations indicate that the low-rank simplicity bias helps when the true function we are trying to approximate is low-rank. On the contrary, if the underlying mapping is a high-rank or the network is made too deep, depth could have a converse effect on generalization. Ample evidence from prior works (Szegedy et al., 2015; He et al., 2016) suggests that over-parameterization of non-linear models improves generalization on fixed datasets, but blindly increasing the number of layers without bells & whistles (e.g., batch-norm, residual connection, etc.) hurts (He et al., 2016).

Fortunately, networks are trained on natural data, where often the goal is to discover a low-rank relationship between the input and the label. Hence, the inductive bias of depth acts as a prior rather than a bug. As noted by Solomonoff (1964) theory of inductive inference, the simplest solution is often the best solution, suggesting that low-rank mapping in neural networks can be used to improve generalization and robustness to overfitting. However, increasing the number of non-linear layers also increases the modeling capacity, thereby making it difficult to isolate the effect of depth.

Nevertheless, since a non-linear network is composed of many linear components, such as fully connected and convolutional layers, we can over-parameterize these linear layers to induce a low-rank bias in the model without increasing the modeling capacity. The details of our linear over-parameterization method are in Appendix C. We observe that such linear over-parameterization improves generalization performance on classification tasks. Furthermore, we find that such implicit regularization outperforms models trained with several choices of explicit regularization. Guo et al. (2020) made a similar empirical observation in the context of model compression where linear over-parameterization improves generalization, but why it works is unexplored.

### 4.1 Image classification with over-parameterization

Using the linear expansion rules in Appendix C, we over-parameterize various architectures and evaluate on a suite of standard image classification datasets: `CIFAR10`, `CIFAR100`, `ImageNet`. All models are trained using SGD with a momentum of 0.9. For data augmentation, we apply a random horizontal flip and random-resized crop. We follow standard training procedures and only modify the network architecture (see Appendix F).

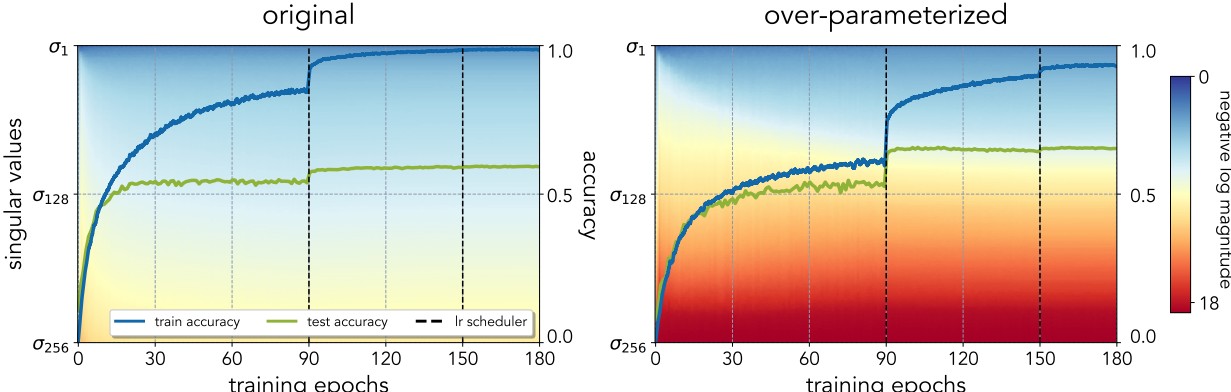

Figure 7: **Training dynamics:** Singular values of the Gram matrix for both original (left) and linearly over-parameterized (right) model throughout training. The models are trained on `CIFAR100` using SGD. Since the first few singular values dominate the distribution, we plot the negative log magnitude of the normalized singular values to better visualize how the intermediate singular values change. The singular values are sorted from largest to smallest $\sigma_i < \sigma_{i+1}$ (top to bottom in the figure) where blue means large and red means small. The original and the over-parameterized models are functionally equivalent and use the same colorbar and scale. The dotted lines ($--$) indicate the learning step schedule, and train and test accuracies are overlayed on top of the distribution. The over-parameterized model learns lower rank embedding and exhibits less overfitting, and has better generalization. See Figure 14 and Figure 15 in the Appendix for the dynamics of the individual weights.

In Figure 7, we compare a CNN trained without (left) and with (right) our over-parameterization (expansion factor $d = 4$) on `CIFAR100`. The CNN consists of 4 convolutional layers and 2 fully connected layers; the architecture details are in Appendix F. We overlay the dynamics of the singular values of the Gram matrix throughout training. The spectral distribution is normalized by the largest singular value and are sorted in descending order $\sigma_i(A) \geq \sigma_{i+1}(A)$ for $i < 1 \leq \min(m, n)$. We observe that both the individual effective weights and the Gram matrix of the over-parameterized model is biased towards low-rank weights. Unlike the original, the majority of the singular values of the over-parameterized model are close to zero. When we take a closer look at the weights of the model, both the original and linearly over-parameterized models first exhibit effective rank contracting behavior throughout training, and then the effective rank starts to increase again – to the best of our knowledge, this is an unexpected training behavior in larger models that are not explained in prior works, possibly because the isometric, balanced initialization, and infinitesimal assumptions made in prior theoretical works do not hold in practice (visualized in Appendix E).

We further quantify the gain in performance from linear over-parameterization in Table 1. The learning rate is tuned per configuration, and we report the best test accuracy throughout the training. We try various over-parameterization configurations and find an expansion factor of 4 to be the sweet spot, with a gain of $+6.3$ for `CIFAR100` and $+2.8$ for `CIFAR10`. The optimal expansion factor depends on the depth of the original network, and in general, we observe a consistent improvement for over-parameterizing models with $< 20$ layers on image classification.

We scale up our experiments to ImageNet, a large-scale dataset consisting of 1.3 million images with 1000 classes, and show that our findings hold in practical settings. For these experiments, we use standardized architectures: AlexNet (Krizhevsky et al., 2012) which consists of 8-layers, and ResNet10 / ResNet18 (He et al., 2016) which consists of 10 and 18 layers, respectively. If our prior observations hold true, we would expect the gain in performance from over-parameterization to be reduced for deeper models. This is, in fact, what we observed in Table 3, with moderate gains in AlexNet and less for ResNet10 and even less for ResNet18. In fact, starting from ResNet34, we observe linearly over-parameterized models perform worse than the original. These experiments support our claim that adding too many layers can over-penalize the model.

| Expansion | | | CIFAR10 | | CIFAR100 | |
|---|---|---|---|---|---|---|
| Factor | FC | Conv | accuracy | gain ↑ | accuracy | gain ↑ |
| ×1 | - | - | 86.9 | - | 57.0 | - |
| ×2 | ✓ | - | 87.1 | +0.2 | 58.4 | +1.4 |
| ×2 | - | ✓ | 87.8 | +0.9 | 61.0 | +4.0 |
| ×2 | ✓ | ✓ | **89.1** | **+2.2** | 61.2 | +4.2 |
| ×4 | ✓ | - | 87.3 | +0.4 | 59.7 | +2.7 |
| ×4 | - | ✓ | **89.1** | **+2.2** | 61.3 | +4.3 |
| ×4 | ✓ | ✓ | 89.0 | +2.1 | **63.5** | **+6.5** |
| ×8 | ✓ | - | 85.9 | -1.0 | 58.8 | +1.8 |
| ×8 | - | ✓ | 88.5 | +1.6 | 61.6 | +4.6 |
| ×8 | ✓ | ✓ | 88.0 | +1.1 | 61.5 | +4.5 |

Table 1: **Over-parameterization ablations:** A nonlinear CNN with 4 convolution and 2 linear layers trained on `CIFAR10` and `CIFAR100` with various degrees of linear over-parameterization. As we have observed with least-squares experiments, there is indeed a sweet spot of depth that is best for generalization. Here we see that linear-overparameterization by 4× performs the best. All models are functionally equivalent and have the same effective number of parameters.

| regularization | CIFAR10 | | CIFAR100 | |
|---|---|---|---|---|
| | accuracy | gain ↑ | accuracy | gain ↑ |
| none (baseline) | 86.9 | - | 57.0 | - |
| low-rank initialization | 86.8 | -0.1 | 57.2 | +0.2 |
| $\ell_2$ norm | 87.2 | +0.3 | 57.0 | +0.0 |
| $\ell_1$ norm | 87.4 | +0.5 | 60.0 | +3.0 |
| nuclear norm | 87.0 | +0.1 | 58.1 | +1.1 |
| effective rank | 86.9 | +0.0 | 57.2 | +0.2 |
| stable rank Sanyal et al. (2019) | 87.6 | +0.9 | 58.3 | +1.3 |
| frobenius$^2$ norm Yoshida & Miyato (2017) | 87.0 | +0.1 | 59.2 | +2.2 |
| over-param (×2) | 89.1 | +2.2 | 61.2 | +4.2 |
| over-param (×2) + $\ell_2$ | 89.6 | +2.7 | 61.1 | +4.1 |
| over-param (×2) + $\ell_1$ | **89.7** | **+2.8** | **63.3** | **+6.3** |

Table 2: **Explicit regularizers:** Comparison of models trained with various regularizers. While explicit low-rank regularizers all result in improved performance, linear over-parameterized deep networks consistently outperform explicit regularizers. The accuracy is computed over the average of 3 runs. Individual runs have $< 0.3\%$ variability in the test performance. All models have the same effective number of parameters.

To find out whether explicit regularizers can approximate the advantages of over-parameterization, we directly compare the performance in Table 2 on `CIFAR`. These regularizers include popular $\ell_1$ and $\ell_2$ norm-based regularizers and commonly-used pseudo-measures of rank. These pseudo-measures of rank, such as *effective rank* and *nuclear norm*, require one to compute the singular value decomposition, which is computationally infeasible on large-scale models. Although we found explicit rank regularizers to help, we observed over-parameterization to outperform models trained with explicit regularizers. Moreover, we found that combining norm-based regularizers with over-parameterization further improves performance. This discrepancy between implicit and explicit regularization may stem from the fact that over-parameterization receives a combined effect of both gradient descent's implicit bias and model parameterization's inductive bias. Therefore, one may need to jointly consider both biases to approximate its effect as an explicit regularizer correctly. Another reason could be that regularizers are inherently different than over-parameterization (Arora et al., 2018). For example, a model trained with a regularizer will have a non-zero gradient, even at zero training loss, while the over-parameterized model will not.

## 5  Discussion

One of the main ingredients in any machine learning algorithm is the choice of hypothesis space: what is the set of functions under consideration for fitting the data? Although this is a critical choice, *how the hypothesis space is also parameterized matters.* Even if two models span the same hypothesis space, the way we parameterize the hypothesis space can ultimately determine which solution the model will converge to – recent work has shown that networks with better neural reparameterizations can find more effective solutions (Hoyer et al., 2019). The automation of finding the right parameterization also has a relationship to neural architecture search (Zoph & Le, 2017), but architecture search typically conflates the search for better hypothesis spaces with the search for better parameterizations of a given hypothesis space. In this work, we have explored just one way of reparameterizing neural nets – stacking linear layers – which does not change the hypothesis space, but many other options exist (see Figure 10 and a short extension to residual networks Appendix I). Understanding the biases induced by these reparameterizations may yield benefits in model analysis and design.

We encourage the readers to look at the appendix for additional experiments and FAQs.

| architecture | ImageNet | | |
|---|---|---|---|
| | original | over-param | gain ↑ |
| AlexNet $[n_\mathsf{layers} = 8]$ (×2) | 57.3 | 59.1 | **+1.8** |
| ResNet10 $[n_\mathsf{layers} = 10]$ (×2) | 62.8 | 63.7 | **+0.9** |
| ResNet18 $[n_\mathsf{layers} = 18]$ (×2) | 67.3 | 67.7 | **+0.4** |

Table 3: **ImageNet:** We show on existing architectures that linear over-parameterization can improve generalization performance. The over-parameterized models have the same number of effective parameters compared to the original. The benefit plateaus when using deeper models. We did not see a noticeable improvement starting from `ResNet34`.

## 6 Related works

**Linear networks**  Linear networks have been used in lieu of non-linear networks for analyzing the generalization capabilities of deep nets. These networks have been widely used for analyzing learning dynamics (Saxe et al., 2014) and forming generalization bounds (Advani et al., 2020). Notable work from (Arora et al., 2018) shows that over-parameterization induces training acceleration which cannot be approximated by an explicit regularizer. Furthermore, (Gunasekar et al., 2017) shows that linear models with gradient descent converge to a minimum nuclear norm solution on matrix factorization. More recently, (Li et al., 2020) demonstrated that gradient descent acts as a greedy rank minimizer in matrix factorization, and (Bartlett et al., 2020; 2021) argues that gradient descent in over-parameterized models leads to benign overfitting. Although mainly used for simplifying theory, (Bell-Kligler et al., 2019) demonstrate the practical applications of deep linear networks.

**Low-rank bias**  Deep linear neural networks have been known to be biased towards low-rank solutions. One of the most widely studied regimes is on matrix factorization with gradient descent under isometric assumptions (Tu et al., 2016; Ma et al., 2018; Li et al., 2018), and further studied on least-squares (Gidel et al., 2019). (Arora et al., 2019a) showed that matrix factorization tends to low nuclear-norm solutions with singular values decaying faster in deeper networks. Complimentary to the analysis of over-parameterization, there has been theoretical work focused on understanding the alignment of gradients in deep networks. Mainly, the works of (Ji & Telgarsky, 2018; 2020) demonstrate that deep networks, under exponential loss, result in low-rank gradients. Note that the aforementioned works focus on why gradient descent finds low-rank solutions. (Pennington et al., 2018) showed that the spectral distribution of the input-output Jacobian is determined by depth. For non-linear networks, understanding the biases has been mostly empirical, with the common theme that over-parameterization of depth or width improves generalization (Neyshabur et al., 2015; Nichani et al., 2020; Golubeva et al., 2021; Hestness et al., 2017; Kaplan et al., 2020). These aforementioned theories have also been adopted for auto-encoding (Jing et al., 2020) and model compression, (Guo et al., 2020). The notion of low-rank bias has some relevance to observations that deep features of similar classes have an inductive bias to be mapped to similar classes Oyallon (2017). More recently, (Pezeshki et al., 2020) have observed that SGD learns to capture statistically dominant features, which leads to learning low-rank solutions, and (Baratin et al., 2021) observed that the alignment of the features acts as an implicit regularizer during training.

**Simplicity bias**  Recent work has indicated that gradient descent in linear models finds max-margin solutions (Soudry et al., 2018; Nacson et al., 2019; Gunasekar et al., 2018). Separately, in the perspective of algorithmic information theory, (Valle-Perez et al., 2019) demonstrated that deep nets' parameter space maps to low-complexity functions. Yang & Salman (2019) extends this observation beyond ReLU networks by analyzing the spectral distribution of the NTK/CK. Furthermore, (Nakkiran et al., 2019b), and (Arpit et al., 2017) have shown that networks learn in stages of increasing complexity. Whether these aspects of simplicity bias are desirable has been studied by (Shah et al., 2020).

**Complexity measures**  A growing number of works have found matrix norm to not be a good measure for characterizing neural networks. (Shah et al., 2018) shows that the minimum norm solution is not guaranteed to generalize well. These findings are echoed by (Razin & Cohen, 2020), which demonstrates that implicit regularization cannot be characterized by norms and proposes rank as an alternative measure.

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

# Appendix

## A   Frequently asked questions

**Q: Why use effective rank?**

**A:** Effective rank was popularized in the deep learning community by Arora et al. (2019a) and has since been a common tool for measuring rank and analyzing the spectral properties of linear layers in neural networks (Razin & Cohen, 2020; Baratin et al., 2021). The entropic definition of the normalized singular values make effective rank a natural measure for computing the effective dimensionality of matrices. The effective rank operates on the distribution of the singular values and not the non-zero counts. Due to numerical imprecisions of modern computation and stochasticity in our algorithms, we often find the rank of the matrix to be full-rank. This requires us to pick a threshold to zero out the smallest singular values after normalization (re-weighting the singular values based on their relative contribution). However, threshold rank is sensitive to the chosen threshold value (see Figure 12) and therefore, we chose effective rank to circumvents these issues.

Alterantive measures, such as nuclear norm, has been commonly used in prior works; but, due to its unbounded nature, it is not invariant to scaling (see Appendix D).

**Q: Does depth always improve generalization?**

**A:** No, we do not claim that over-parameterization will always improve generalization. Like any regularizers, over-regularizing your model hurts performance, as we have seen with linear models that are made too deep (He et al., 2016). When the true underlying function is low-rank (as is typically with natural data), the bias toward low-rank kernels is beneficial as training will tend to find a good fit that also matches the true structure of the data (and therefore generalizes well). When the true function is not low-rank, the bias will have an adverse effect (see Figure 12).

**Q: Are comparisons made at comparable loss value?**

**A:** Yes, our experiments either assume the models have reached zero training error or have the same modeling capacity.

**Q: What is the contribution of work?**

**A:** While there is ample evidence that low nuclear norm bias exists in over-parameterized models (Gunasekar et al., 2017; Arora et al., 2019a; Li et al., 2020), these theoretical works make assumptions that com-promise practical insights for grounded mathematical explanation (see related works). These assumptions are used to derive theoretical guarantees and often require: linear assumptions, infinite width networks, dynamical isometries, gradient flow dynamic, or a specific learning paradigm such as matrix completion. In addition, low nuclear norm bias is not necessarily the same as low-rank bias, in which low-rank bias is more closely knit to the spectral bias. Hence, it is unclear whether nuclear norm is the only explanation for the low-rank bias and whether these explanations hold true in practice. In light of empiricism, we provide a series of investigations on the role of over-parameterization with the hopes to better guide our theoretical and practical understanding of deep networks.

To our knowledge, our work is the first to extensively study the existence of low-rank bias in non-linear networks. We extend our observations to practical learning paradigms. We show that the low-rank bias exists even before and after training, and highlight that gradient descent is not the sole explanation.

**Q: How does our work differ with Arora et al. (2019a)?**

Arora et al. (2019a) study the implicit bias of gradient descent in over-parameterized models. Contributions of their work include:

- Extends the conjecture of Gunasekar et al. (2017) that gradient descent in linear matrix factorization results in low nuclear norm solution.
- The observation is that deeper linear models can better solve low-rank matrix factorization problems using gradient descent on toy tasks.
- Provides theory, under isometric assumptions, how depth plays a role in the learning dynamics of gradient descent in linear networks. "The dynamics promote solutions that have a few large singular values and many small ones, with a gap that is more extreme the deeper the matrix factorization is".

Our work provides new insights in the role of over-parameterization in many ways. We highlight few of these differences below:

- We extend the observation on simplicity bias to linear and non-linear finite networks. Our observations do not depend on any isometric assumptions, and we show that it holds even in practical problem setups.
- We show that the parameterization bias exists regardless of training. That is, even with

or without optimization, models are biased towards low effective rank mappings.

- We show that implicit bias exists beyond gradient descent, whereas prior theory only applied to models optimized with gradient descent.
- Even at zero-training error, models with different depths but the same capacity will converge to different solutions, and ultimately exhibit different generalization properties.
- We show that even in practical learning paradigms, such as classification on CIFAR10 and ImageNet, deep networks exhibit the low effective rank bias.

In summary: while prior works have primarily studied the bias of over-parameterization under the context of gradient descent, our work focuses on bringing attention to another missing piece in understanding why over-parameterized models converge to low-rank solutions – a phenomenon which is often credited to why deep networks generalize (Gunasekar et al., 2017; Arora et al., 2019a; Razin & Cohen, 2020).

**Q: Is the low-rank phenomena a trivial observation?** While this is true and well-known for linear models with discrete rank, our work is making a more subtle statement that the spectral distribution of the weights becomes more concentrated as models are made deeper — the entropy of the spectral distribution decreases. In addition, our empirical findings are not directly predicted by the fact that multiplying matrices reduces rank: the networks also exhibit this effective rank reducing behavior at initialization and also at convergence. It is also unknown whether these behaviors would still persist for non-linear networks.

**Q: Could the benefits of the proposed over-parametrization might be just due to increased capacity?** Throughout our paper, we demonstrated that even when the model does not have increased capacity, linearly over-parameterized models improve generalization (See Figure 6, Figure 7, Table 1, Table 3, Table 2). Furthermore, even when all models achieve the same training error, we demonstrated that the resulting generalization properties are different (See Figure 12). We showed empirically that that model parameterization ultimately determines the likelihood of the hypothesis space – deep models put higher probability weight on lower effective rank embedding.

**Q: What are the standard deviation on these classification experiments?** In supervised classifications, the standard deviations are very small. All the experiments in our work have less than $< 0.3\%$

standard deviation. For the sake of making the tables readable, we have decided to omit the standard deviations in the tables.

**Q: What is the relevance of analyzing the gram matrix in non-linear model?**

The relevance of analyzing the gram matrix in non-linear models is not straightforward. This is because depth with non-linear layers can increase functional expressivity while also decreasing the rank. Hence, we should consider comparing models' gram-matrix when either the models have the same functional power or when the models that are being compared are operating in the zero-training error regime.

There are many reasons why one would want to analyze the gram matrix in non-linear models (Montavon et al., 2011). Under the conditions of functional equivalence or zero-training error, one can make relative comparisons on how the data is being mapped on held out data. In natural data, it is often assumed that we are trying to discover a low-rank relationship between the input and the label. For example, a model that overfits to every training sample without inferring any structure on the data will generally have a test gram-matrix that is higher rank than that of a model that has learned parsimonious representations. Furthermore, the low-rank gram matrix is also a good indicator of the variability in the data mapping. Lower rank on held out data indicates less excess variability and therefore could be a good for analyzing robustness.

**Q: Relationship to infinitely wide networks?**

Analyzing the spectral properties of deep networks has also been studied under infinite width neural networks. (Aitchison et al., 2021) have observed that deep kernel processes with fixed, non-learned kernels exhibit a lower-rank structure, where the kernel follows power-law structure with depth. (Yang & Salman, 2019) shows that NTKs also exhibit this simplicity bias. For Gaussian processes, (Aitchison, 2020; Zavatone-Veth et al., 2021) further demonstrates that the rank of the "output Gram matrix" is restricted to the dimensionality of the output space.

# B   Random matrix theory in finite models

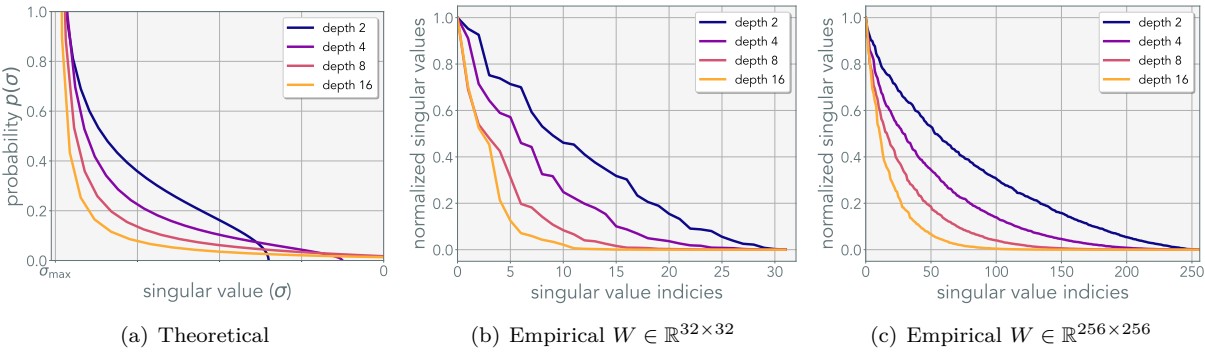

(a) Theoretical      (b) Empirical $W \in \mathbb{R}^{32 \times 32}$      (c) Empirical $W \in \mathbb{R}^{256 \times 256}$

Figure 8: **Theoretical and empirical singular-value distributions:** We show that even on finite matrices, the singular-value distribution matches that of the theoretical distribution. This implies that deeper finite-width linear neural networks should have lower effective rank in practice. The theoretical distribution uses an unnormalized probability distribution.

Random matrix theory makes an infinitely large random matrix assumption (square or rectangular); one can think of them as infinitely wide neural networks. This infinitely large matrix assumption is used to derive a deterministic spectral distribution (singular-value distribution) of random matrices. In Figure 8, we show that the empirical spectral distribution closely follows that of the theoretical distribution derived in (Pennington et al., 2017; Neuschel, 2014). Even when using a very small weight matrix of size $W \in \mathbb{R}^{32 \times 32}$, and more so on larger weight matrices $W \in \mathbb{R}^{256 \times 256}$, the singular values are dominated by just a few values when increasing the number of layers.

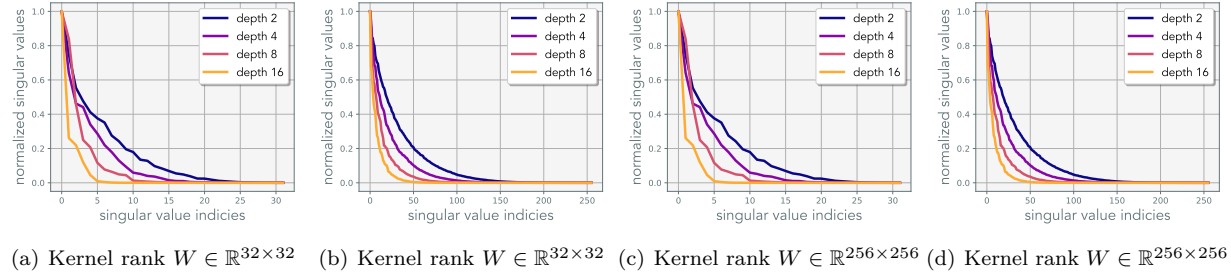

(a) Kernel rank $W \in \mathbb{R}^{32 \times 32}$    (b) Kernel rank $W \in \mathbb{R}^{32 \times 32}$    (c) Kernel rank $W \in \mathbb{R}^{256 \times 256}$    (d) Kernel rank $W \in \mathbb{R}^{256 \times 256}$

Figure 9: **Singular value distribution of Gram matrices:** Similar to the singular-value distribution of the weights, the singular-value distribution of gram matrices also become sharper, lower effective rank, with increased depth.

In a similar light, we can also empirically observe the gram matrices' spectral distribution. As shown in Figure 9, we observed that gram matrices also exhibit almost the same trend predicted by random matrix theory. It is natural to assume that the theory has no practical meaning when the networks are trained, and the weight matrices are no longer random. Hence we trained the models to convergence on least-squares objective and observed the spectral distribution to maintain its depth-wise separation as observed during initialization. These observations help reaffirm our conjecture and further motivate the potential usefulness of random matrix theory in understanding the role of over-parameterization in deep networks.

## C Expanding a non-linear network

A deep non-linear neural network with $l$ layers is parameterized by a set of $l$ weights $W = \{W_1, \ldots, W_l\}$. The output of the $j$-th layer is defined as $\phi_j = \psi(f_{W_j}(\phi_{j-1}))$, for some non-linear function $\psi$ and input feature $\phi_{j-1}$. The initial feature map is the input $\phi_0 = x$, and the output is the final feature map $y = \phi_l$. We can expand a model by depth $d$ by expanding all linear layers, i.e. redefining $f_{W_j} \to f_{W_j^d} \circ \cdots \circ f_{W_j^1} \ \forall \ j \in \{1, ..., l\}$. We illustrate this in Figure 10. We describe this operation for fully connected and convolutional layers.

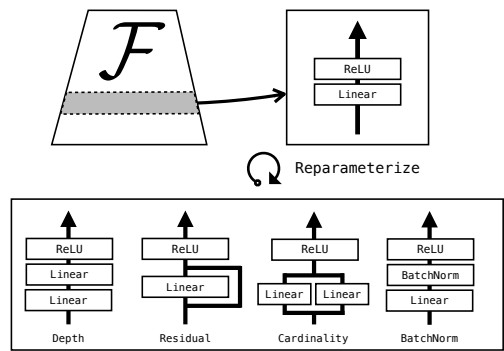

Figure 10: **Linear reparameterization**: For a model $\mathcal{F}$, we can reparameterize any linear layer to another functionally equivalent layer (shown in the box below). In this work we mainly explore reparameterization of depth. Batch-norm and any other running-statistics driven normalization layers are linear only at test time.

**Fully-connected layer** A fully-connected layer is parameterized by weight $W \in \mathbb{R}^{m \times n}$. One can over-parameterize $W$ as a series of linear operators defined as $\prod_{i=1}^{d} W_i$. For example, when $d = 2$, $W \to W_2 W_1$, where $W_2 \in \mathbb{R}^{m \times h}$ and $W_1 \in \mathbb{R}^{h \times n}$ for some hidden dimension $h$. The variable $h$ is referred to as the width of the expansion and can be arbitrarily chosen. In our experiments, we choose $h = n$ unless stated otherwise. Note that $h < \min(m, n)$ would result in a rank bottleneck and explicitly reduce the underlying rank of the network.

**Convolutional layer** A convolutional layer is parameterized by weight $W \in \mathbb{R}^{m \times n \times k \times k}$, where $m$ and $n$ are the output and input channels, respectively, and $k$ is the dimensionality of the convolution kernel. For convenience, we over-parameterize by adding $1 \times 1$ convolution operations. $W_d * W_{d-1} * \cdots * W_1$, where $W_d \in \mathbb{R}^{m \times h \times 1 \times 1}$, $W_{d-1}, ..., W_2 \in \mathbb{R}^{h \times h \times 1 \times 1}$ and $W_1 \in \mathbb{R}^{h \times n \times k \times k}$. Analogous to the fully-connected layer, we choose $h = n$ to avoid rank bottleneck.

The work by Golubeva et al. (2021) explores the impact of width $h$. Similar to their findings, we observed using the larger expansion width to slightly improve performance. We use $h = 2n$ for our ImageNet experiments.

# D    Comparisons of rank measures and kernel distance functions

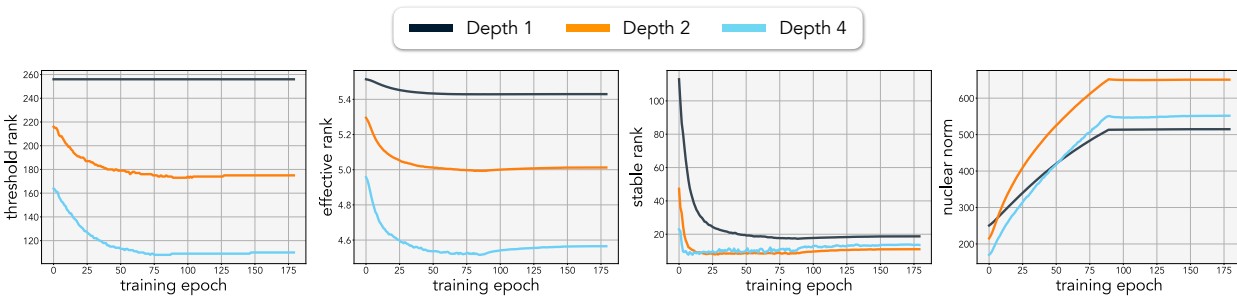

Figure 11: **Comparing rank-measures:** Comparison between various pseudo-metrics of rank when varying the number of layers. The threshold is set to $\tau = 0.01$ for threshold rank.

The rank of a matrix – which defines the number of independent basis – in practice can often be a sub-optimal measure. For deep learning, fluctuations in stochastic gradient descent and numerical imprecision can easily introduce noise that causes a matrix to be full-rank. In addition, simply counting the number of non-zero singular values may not indicate what we care about in practice: the relative impact of the $i$-th basis compared to the $j$-th basis. In a typical image classification setup, we observed that the norm of the matrix often increases during training. This is highlighted by the nuclear norm in Figure 11. Coupled with numerical imprecisions, we found that the weights of the matrix are often always full rank.

A rank measure closest to the true definition of rank would be thresholded-rank, where the smallest singular values are thresholded after normalization (re-weighting the singular values based on relative contribution). However, thresholded rank is very sensitive to the threshold value one chooses (shown below); hence we used effective rank to avoid this issue.

**Definition D.1** (Effective rank). *(Roy & Vetterli, 2007)*

*For any matrix $A \in \mathbb{R}^{m \times n}$, the effective rank $\rho$ is defined as the Shannon entropy of the normalized singular values:*

$$\rho(A) = - \sum_{i=1}^{\min(n,m)} \bar{\sigma}_i \log(\bar{\sigma}_i),$$

*where $\bar{\sigma}_i = \sigma_i / \sum_j \sigma_j$ are the normalized singular values such that $\sum_i \bar{\sigma}_i = 1$. It follows that $\rho(A) \leq \operatorname{rank}(A)$. This measure is also known as the spectral entropy.*

The effective rank has been previously used as a surrogate measure for measuring the rank of neural network weights ((Arora et al., 2019a)). We now state other various metrics that have been used as a pseudo-measure of matrix rank. One obvious alternative is to use the original definition of rank after normalization:

**Definition D.2** (Threshold rank). *For any matrix $A \in \mathbb{R}^{m \times n}$, the threshold rank $\tau$-Rank is the count of non-small singular values after normalization:*

$$\tau\text{-Rank}(A) = \sum_{i=1}^{\min(n,m)} \mathbb{1}[\bar{\sigma}_i \geq \tau],$$

*where $\mathbb{1}$ is the indicator function, and $\tau \in [0,1)$ is the threshold value. $\bar{\sigma}_i$ are the normalized singular values defined above.*

It is worth noting that not normalizing the singular values results in the numerical definition of rank. As stated before, the threshold rank depends largely on the threshold value and therefore a drastically different scalar representation of rank can emerge. Potentially, a better usage of threshold rank is to measure the AUC when varying the threshold.

Related to the definition of the threshold rank, stable rank operates on the normalized squared-singular values:

**Definition D.3** (Stable rank). *(Vershynin, 2018)*

*For any matrix, $A \in \mathbb{R}^{m \times n}$, the stable rank is defined as:*

$$\mathsf{SRank}(A) = \frac{\|A\|_F^2}{\|A\|_2} = \frac{\sum \sigma_i^2}{\sigma_{max}^2},$$

*Where $\sigma_i$ are the singular values of $A$.*

Stable-rank provides the benefit of being efficient to approximate via the power iteration (Mises & Pollaczek-Geiringer, 1929). In general, stable-rank is a good proxy for measuring the rank of the matrix and has been used in prior works such as (Nichani et al., 2020). This is not necessarily true when the singular values have a long tail distribution, which under-emphasizes the small singular values un-proportionately due to the squared-operator. We observed that the largest singular values often get over exaggerated in neural networks and hence we often found that $\mathsf{SRrank}$ converges to values close to 1, making insightful observations impractical.

Lastly, the nuclear norm has been considered as the de facto measure of rank for the task of matrix factorization/completion, with low nuclear-norm indicating that the matrix is low-rank:

**Definition D.4** (Nuclear norm). *For any matrix $A \in \mathbb{R}^{m \times n}$, the nuclear norm operator is defined as:*

$$\|A\|_* = \text{tr}(\sqrt{AA^T}) = \sum_{i}^{\min(n,m)} \sigma_i(A)$$

*Where $\sigma_i$ are the singular values of $A$.*

Nuclear norm, however, has obvious flaws of being an un-normalized measure. The nuclear norm is dictated by the magnitude of the singular values and not the ratios. Therefore, the nuclear norm can be made arbitrarily large or small without changing the output distribution.

The comparisons of these metrics are illustrated in Figure 11 where effective rank has the closest behavior to that of the thresholded rank. The metrics are computed on the end-to-end weights throughout the training. We use linear over-parameterized models with various depths on least-squares.

In Figure 12, we repeat our least-squares experiments from our main paper using thresholded rank with various threshold values $\tau = \{0.001, 0.005, 0.01\}$. We show that the effective rank indeed correlates well with the thresholded rank. As stated above, we observe that the rank drastically changes depending on the threshold value. We also run the same experiment on varying task-ranks of 30, 16, and 4. Although all models span the same set of functions (same effective weight dimensionality), the resulting generalization performance differs depending on the depth of the model. In a high task-rank setting, the generalization error increases with depth, while generalization error decreases with depth in a low task-rank setting. This indicates the parameterization of the model determines the hypothesis space the model explores during training, which aligns with our conjecture and our observations. This is further highlighted in medium and low task-rank settings, where all models reach zero-training error, yet the test-loss differs.

**Kernel distance functions** In our work we used cosine kernels to construct the Gram matrices. Cosine kernels are normalized linear kernels and we found it to produce cleaner results. Cosine kernels has been commonly used as distance function to measure similarity between features (Zhang et al., 2018). We further show in Figure 13 that Gram matrices constructed with kernel distance functions such as linear kernels and correlation kernels also exhibit the low-rank simplicity bias.

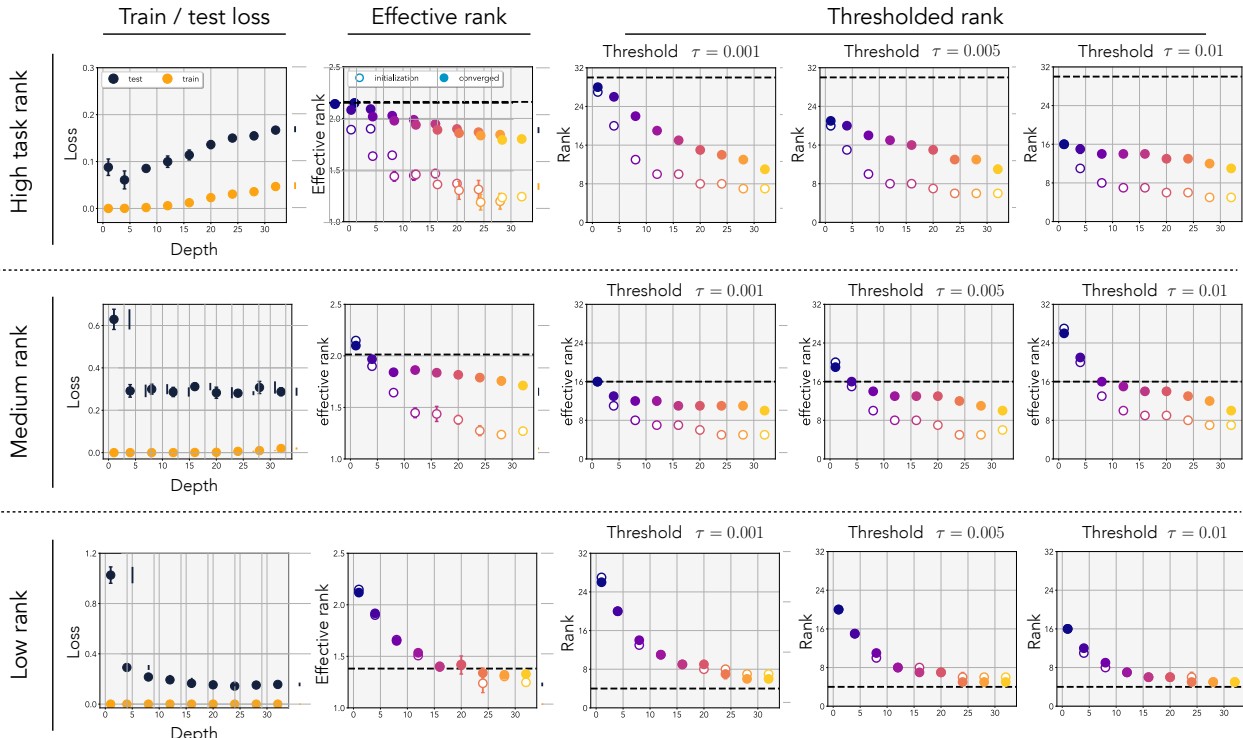

Figure 12: **Least-sqaures ablation**: Least-squares experiment using both effective-rank and thresholded rank measure. We run the experiments on various task-ranks $30, 16, 4$. For thresholded rank, we use various threshold values of $\tau = \{0.001, 0.005, 0.01\}$ and show that it correlates well with effective rank. The thresholded rank has a downside of being sensitive to the threshold values, and one has to subjectively tune the suitable threshold, making it a suboptimal choice. The figure shows that depending on the rank of the task, the generalization performance depends on the depth. When the task rank is high, shallower models perform better, and when the task rank is low, deeper models perform better. This aligns with our observation that the model parameterization biases the hypothesis search space in neural networks even if the models are effectively the same and span the same set of functions.

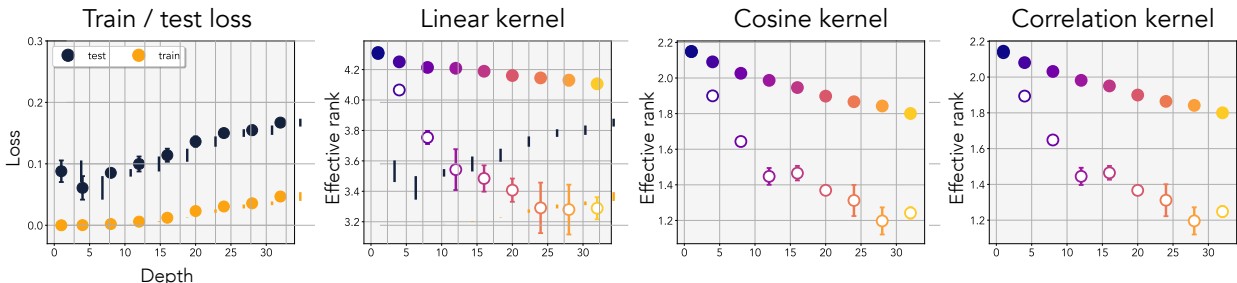

Figure 13: **Kernel ablation**: We ablate our least-squares experiments by using various kernel distance functions. Cosine kernels are normalized version of linear kernels, and pearseon-correlation kernels are another way of normalizing linear kernels. We can see that all kernels show the same behavior.

# E  Singular value dynamics of weights

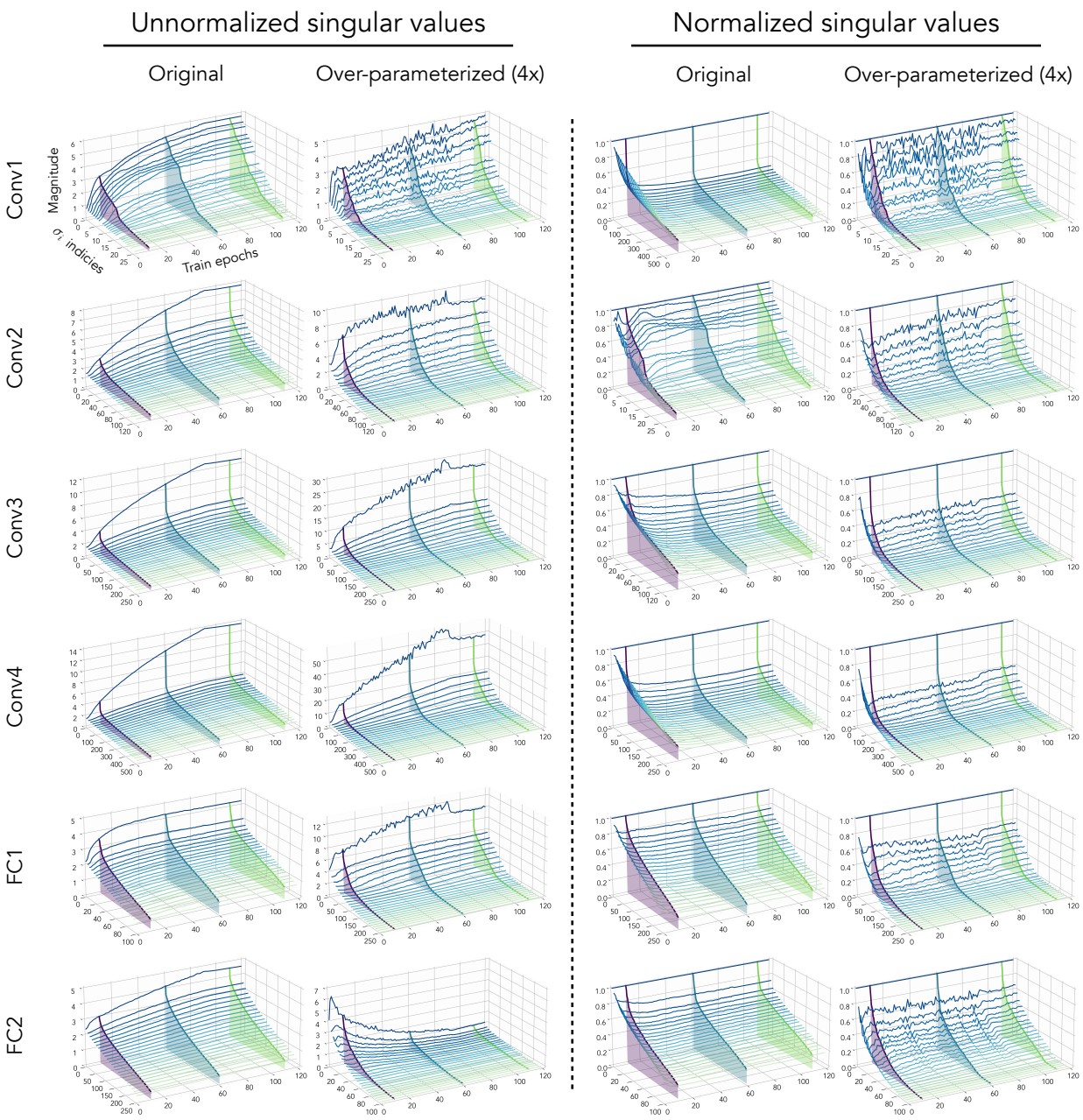

Figure 14: **Dynamics per layer**: The singular values of the individual weights during `CIFAR100` training. On the left we have the unnormalized singular values, and on the right the distributions are scaled by the largest singular values. We uniformly subsample 24 singular values for the visualization. The cross sections are provided to help visualize the distribution at that specific epoch. The individual lines track the singular values $\sigma_i$ over time.

In Figure 14, we visualize the singular values of the individual weights when training on `CIFAR100` image classification for the first 120 epochs. The cross-sections indicate the singular value distribution at that specific epochs. For the over-parameterized model, the effective rank is computed on the effective weight. On the left, we plot the unnormalized singular values and observe that the norm of the singular values

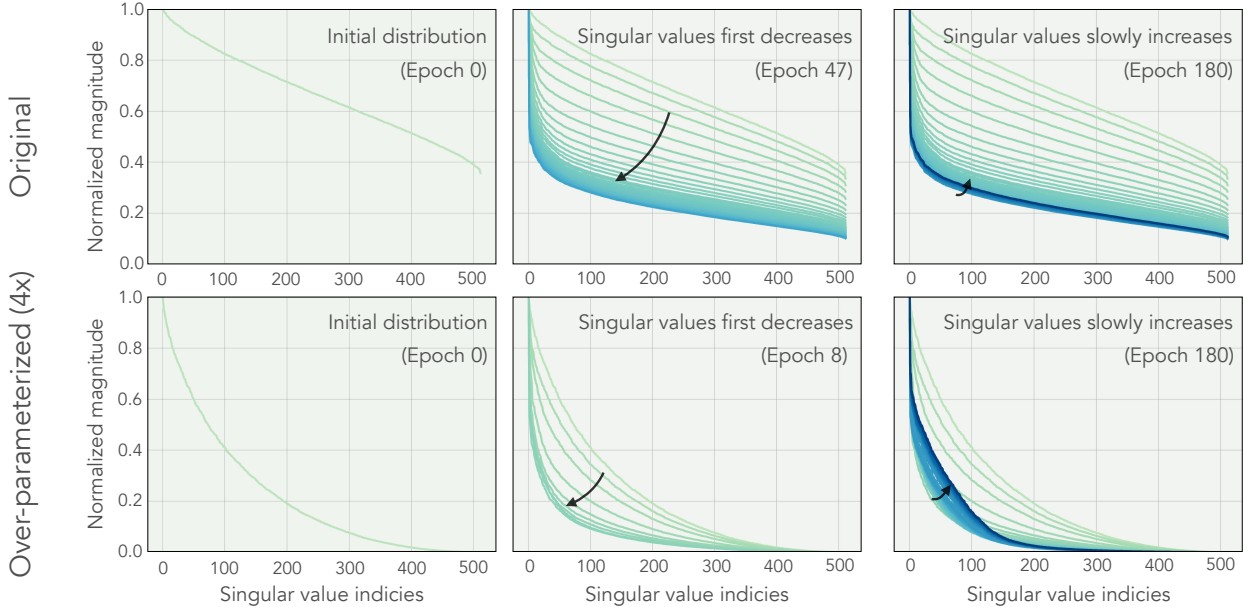

Figure 15: **Dynamics overlay**: We overlay the singular values of the `Conv4` weights. We observed that the effective rank first rapidly decreases early on in the training and then bounces back up slowly throughout the rest.

increases throughout training for all layers except for the last classification layer in the over-parameterized model. When we normalized the distribution by the largest singular value $\sigma_0$ (right), we observed that the distribution becomes sharper early in training but does not change much throughout.

To get a better sense of how the distribution evolves over time by overlaying the distribution on top of each other. In Figure 15, we overlay the distribution on top of each other for `Conv4` weights and observed that the effective-rank first decays rapidly and then slightly increases throughout the rest of the training. This dynamical behavior, to our knowledge, is not explained in prior theoretical works and could highlight the dissonance between the assumptions made in theory do not fully describe behaviors observed in practice.

## F  Training details and model architecture

For Figure 1, we trained a ReLU network with input, output, and the hidden dimension of 64; the larger the width, the more pronounced the effects seemed to be. We chose 64 due to the run time of these models. We train the model using SGD with a momentum of 0.9, and we do not use weight decay. We observed that very deep networks become very sensitive to the learning rate. Therefore, we tuned the learning rate per model. For each model we trained using the learning rates $[1.0, 0.5, 0.2, 0.1, 0.05, 0.02, 0.01, 0.005, 0.002, 0.001]$ and chose the best performing learning rate. A heuristics we found somewhat helpful is setting the learning rate as $\eta \propto \frac{1}{\sqrt{d}}$ for some depth $d$. The weights are initialized using normal distribution and linearly swept through the scale of the variance. We found the gain of $\sqrt{2}$, the default gain of Kaiming initialization (He et al., 2016), to work the best. We tried 5 different seeds for each task-rank and also 5 different initialization seed for the neural network and observed a consistent result. All models are trained for 24000 epochs as we observed deeper models take a long time to converge. For shallower models, it is sufficient to train them for roughly 1000 epochs. We also experimented with learning rate schedulers but only helped a little. For all models, we step the learning rate by a factor of 10 at epoch 18000. For all experiments $\text{rank}(W^*) = \{1, 4, 16, 32, 64\}$, we use total of 128 training samples. Using a different number of training samples results in similar observations. We experimented with both SGD and GD and observed the same phenomena. For SGD, we used a mini-batch size of 32. **When the rank of the underlying function is high, we found that it required significantly more fine-grained tuning of the hyper-parameters.**

All models for image classification are trained using `PyTorch` (Paszke et al., 2019) with `RTX 2080Ti` GPUs. We use stochastic gradient descent with a momentum of 0.9. For CIFAR experiments, the initial learning rate is individually tuned (0.02 for most cases), and we train the model for 180 epochs. We use a step learning rate scheduler at epoch 90 and 150, decreasing the learning rate by a factor of 10 each step. For all the models, we use random-horizontal-flip and random-resize-crop for data augmentation.

The training details for ImageNet can be found in https://github.com/pytorch/examples/blob/master/imagenet. When linearly over-parameterizing our models, we bound the variance of the weights using Kaiming initialization (He et al., 2016), a scaled Normal distribution. This allows us to have the same output variance, regardless of the number of layers we over-parameterize our models by. We found this to be critical for stabilizing our training. We also found it important to re-tune the weight decay for larger models on ImageNet. The architecture used for the `CIFAR` experiments is:

| CIFAR architecture |
|:---:|
| RGB image $y \in \mathbb{R}^{32 \times 32 \times 3}$ |
| Convolution $3 \rightarrow 64$, MaxPool, ReLU |
| Convolution $64 \rightarrow 128$, MaxPool, ReLU |
| Convolution $128 \rightarrow 256$, MaxPool, ReLU |
| Convolution $256 \rightarrow 512$, ReLU |
| GlobalMaxPool |
| Fully-Connected $512 \rightarrow 256$, ReLU |
| Fully-Connected $256 \rightarrow$ num classes |

We tuned the learning rate per model as deeper models (8x expansion or more) become sensitive to the initial learning rate. This was critical for the least-squares experiments but not so much for `CIFAR` and `ImageNet` experiments (since we used up to 8x expansion). The one hyper-parameter that we found that needed tuning was the weight decay in `ImageNet` classification. A typical 2x or 4x expansion does not require much tuning

at all. The learning rate scheduler was originally tuned to the baseline and was held fixed. The learning rate decay for baselines with explicit regularizers was tuned.

For least-squares experiments, we were unable to achieve zero-training error for very deep networks using various common optimization tricks. We argue that the parametric bias of depth is the reason why these models are unable to overfit to high-rank data. While it is certainly possible that an optimizer or optimization setting would allow us to reach zero-training error, we were unable to find such a setting by sweeping across hyper-parameters and common optimization techniques. Given an SGD optimizer, we tuned learning rates ($\{0.1, 0.05, 0.01, 0.005, 0.001\}$), momentum (($\{0.1, 0.5, 0.9, 0.99\}$), learning rate schedulers ($\{\mathsf{none}, \mathsf{step}, \mathsf{decay\ on\ plateau}, \mathsf{cosine}\}$). We found the best set of hyperparameters that minimizes the training loss is with momentum set to 0.9 and using $\mathsf{decay\ plateau}$ scheduler. For an over-parameterized linear network of depth 16, and underlying task rank set to 24, we show that even with the best set of hyper-parameters, the training loss cannot be perfectly minimized:

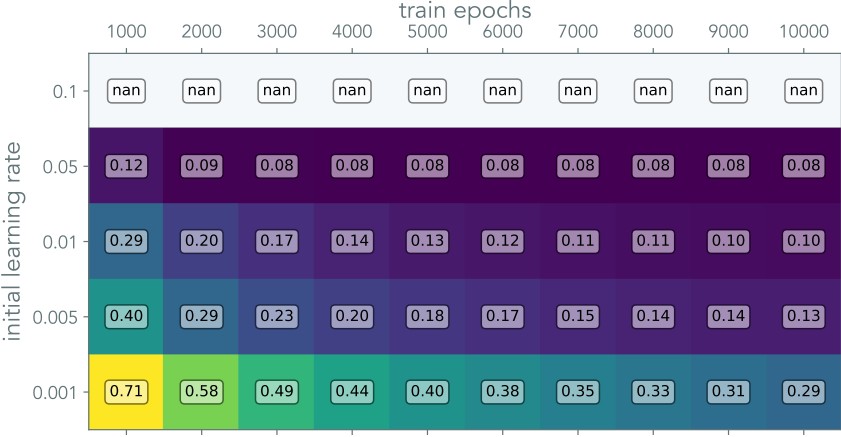

Figure 16: **Training error vs learning-rate:** Training error with varying learning rates for least-squares trained on a linear network with a depth of 24.

# G   Differential effective rank

To analyze the effective rank as a function of the number of layers, we define a differential variant of the effective rank. This formulation allows us to use the fact that the eigen/singular-value spectrum assumes a probability distribution in the asymptotic case.

**Definition G.1** (Differential effective rank). *For any matrix $A \in \mathbb{R}^{m \times n}$ as $\min(m, n) \to \infty$ the singular values assume a probability distribution $p(\sigma)$. Then, we define the differential effective rank $\rho$ as:*

$$\rho(A) = -\int_0^{\sigma_{\max}} \frac{\sigma}{c} \log(\frac{\sigma}{c}) p(\sigma) d\sigma \tag{4}$$

*where $p(\sigma)$ is the singular value density function and $c = \int_0^{\sigma_{\max}} \sigma p(\sigma) d\sigma$ is the normalization constant.*

# H   Proof of Theorem 1

To prove Theorem 3.1, we leverage the findings from random matrix theory, where the singular values assume a probability density function. Specifically, we use the density function corresponding to the singular values of the matrix $W$ composed of the product of $L$ individual matrices $W = W_L, \ldots W_1$, where the components of the matrices $W_1$ to $W_L$ are drawn i.i.d from a Gaussian. Characterizing such density function is, in general intractable, or otherwise very difficult. However, in the asymptotic case where $dim(W) \to \infty$ and $W$ is square, the density function admits the following concise closed-form (Eq. 13 of Pennington et al. (2017) derived from Neuschel (2014)):

$$p(\sigma(\phi)) = \frac{2}{\pi} \sqrt{\frac{\sin^3(\phi) \sin^{L-2}(L\phi)}{\sin^{L-1}((L+1)\phi)}} \quad \sigma(\phi) = \sqrt{\frac{\sin^{L+1}((L+1)\phi)}{\sin(\phi) \sin^L(L\phi)}}, \tag{5}$$

where $\sigma$ denotes singular values (parameterized by $\phi \in [0, \frac{\pi}{L+1}]$) and $p$ denotes the probability density function of $\sigma$ for $\sigma \in [0, \sigma_{\max}]$, and $\sigma_{\max}^2 = L^{-L}(L+1)^{L+1}$. The parametric probability density function spans the whole singular value spectrum when sweeping the variable $\phi$.

We are interested in computing the effective rank of $W$. Using the above density function, we can write it in the form:

$$\rho(W) = -\int_0^{\sigma_{\max}} \frac{\sigma}{c} \log(\frac{\sigma}{c}) p(\sigma) d\sigma, \tag{6}$$

We now write this integral in terms of $\phi$ as the integration variable, such that we can leverage the density function in Eqn. 5. Using the change of variable, we have:

$$\rho(W; L) = -\int_0^{\frac{\pi}{L+1}} \frac{\sigma(\phi)}{c} \log(\frac{\sigma(\phi)}{c}) \big(-p(\sigma(\phi))\sigma'(\phi)\big) d\phi, \tag{7}$$

where $\sigma'(\phi) = \frac{d}{d\phi}\sigma(\phi)$. Note that the integral limits $[0, \sigma_{\max}]$ on $\sigma$ respectively translate[1] into $[\frac{\pi}{L+1}, 0]$ on $\phi$, where,

$$-p(\sigma(\phi))\sigma'(\phi) = \frac{1}{2\pi}\Big(1 + L + L^2 - L(L+1)\cos(2\phi) - (L+1)\cos(2L\phi) + L\cos(2(1+L)\phi)\Big) \csc^2(L\phi).$$

In the following, we treat $L$ as a continuous variable, and show that $\rho(W; L)$ is decreasing in $L$. This is sufficient for proving $\rho(W; L)$ results in a decreasing sequence at integer values of $L$.

As $\rho(W; L)$ is differentiable in $L$, $\rho(W; L)$ decreases in $L$ if and only if $\frac{d\rho}{dL} < 0$. Since integration and differentiation are w.r.t. different variables, they commute; we can first compute the derivative of the integrand w.r.t. $L$ and then integrate w.r.t. $\phi$ and show that the result is negative.

---

[1]note that the direction of integration needs to flip (by multiplying by -1) to account for flip of the upper and lower limits.

With abuse of notation, we rewrite Eqn. 7 by making the dependency of functions on $L$ explicit.

$$\rho(W; L) \quad = \quad \int_0^{\frac{\pi}{L+1}} \frac{\sigma(\phi, L)}{c(L)} \log(\frac{\sigma(\phi, L)}{c(L)}) p(\sigma(\phi, L)) \sigma'(\phi, L) \, d\phi \,, \tag{8}$$

where $\sigma'(\,.\,,\,.\,)$ denotes partial derivative of $\sigma(\,.\,,\,.\,)$ w.r.t. its first argument.

We now proceed with differentiating $\rho$ w.r.t. $L$. Notice that, besides the integrand, the integral limit depends on $L$ as well. Thus can be handled using Leibniz integral rule for differentiation, which yields,

$$\frac{\partial \rho}{\partial L} = \quad \left( \frac{\sigma(\phi, L)}{c(L)} \log(\frac{\sigma(\phi, L)}{c(L)}) p(\sigma(\phi, L)) \sigma'(\phi, L) \right)_{\phi \to \frac{\pi}{L+1}} \left( \frac{\partial}{\partial L} \frac{\pi}{L+1} \right) \tag{9}$$

$$+ \quad \int_0^{\frac{\pi}{L+1}} \frac{\partial}{\partial L} \left( \frac{\sigma(\phi, L)}{c(L)} \log(\frac{\sigma(\phi, L)}{c(L)}) p(\sigma(\phi, L)) \sigma'(\phi, L) \right) d\phi \tag{10}$$

It is easy to verify that,

$$\lim_{\phi \to \frac{\pi}{L+1}} \frac{\sigma(\phi, L)}{c(L)} \log(\frac{\sigma(\phi, L)}{c(L)}) \quad = \quad 0 \tag{11}$$

$$\lim_{\phi \to \frac{\pi}{L+1}} p(\sigma(\phi, L)) \sigma'(\phi, L) \quad = \quad \frac{(L+1)\left(L \cos(\frac{2\pi}{1+L}) + \cos(\frac{2L\pi}{1+L}) - 1 - L\right) \csc^2(\frac{L\pi}{1+L})}{2\pi} \tag{12}$$

Consequently,

$$\left( \frac{\sigma(\phi, L)}{c(L)} \log(\frac{\sigma(\phi, L)}{c(L)}) p(\sigma(\phi, L)) \sigma'(\phi, L) \right)_{\phi \to \frac{\pi}{L+1}} = 0 \,. \tag{13}$$

This allows us to drop the first term in $\frac{\partial \rho}{\partial L}$ to express it more compactly as,

$$\frac{\partial \rho}{\partial L} \quad = \quad \int_0^{\frac{\pi}{L+1}} \frac{\partial}{\partial L} \left( \frac{\sigma(\phi, L)}{c(L)} \log(\frac{\sigma(\phi, L)}{c(L)}) p(\sigma(\phi, L)) \sigma'(\phi, L) \right) d\phi \,. \tag{14}$$

It is messy but straightforward to compute $\frac{\partial}{\partial L} \left( \frac{\sigma(\phi,L)}{c(L)} \log(\frac{\sigma(\phi,L)}{c(L)}) p(\sigma(\phi, L)) \sigma'(\phi, L) \right)$. Integrating that w.r.t. $\phi$ from 0 to $\frac{\pi}{L+1}$ leads to a negative expression, thus $\frac{\partial \rho}{\partial L} < 0$.

The proof here considers the asymptotic case when $\dim(W) \to \infty$. This limit case allowed us to use the probability distribution of the singular values. Although we do not provide proof for the finite case, our results demonstrate that it holds empirically in practice (see Figure 2).

# I Extension to residual connections

This work concentrates our analysis on depth and its role in both linear and non-linear networks. Yet, the ingredients that make up what we know as state-of-the-art models today are more than just depth. From cardinality (Xie et al., 2017) to normalization (Ioffe & Szegedy, 2015) and residual connections (He et al., 2016), numerous facets of parameterization have become a fundamental recipe for a successful model (see Figure 10). Of these, residual connections have the closest relevance to our work.

What is it about residual connections that allow the model to scale arbitrarily in depth? while vanilla feed-forward networks cannot? One possibility is that beyond a certain depth, the rank of the solution space reduces so much that good solutions no longer exist. In other words, the implicit rank-regularization of depth may take priority over the fit to training data. Residual connections are essentially "skip connections" that can be expressed as $W \rightarrow W + \mathbf{I}$, where $\mathbf{I}$ is the identity matrix (Dirac tensor for convolutions). There are two interpretations of what these connections do: one is that identity preservation

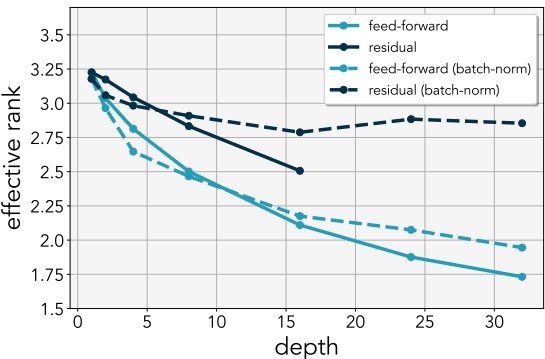

Figure 17: **Residual connections:** The effective rank of linear models trained with and without residual connection on a low-rank least-squares problem. Contrary to feed-forward networks, residual networks maintains the effective rank of the weights even when adding more layers. Residual networks without batch-normalization suffer from unstable output variance after 16 layers.

prevents the rank-collapse of the solution space. The other interpretation is that residual connections reduce the *effective depth* — the number of linear operators from the input to the output (e.g., ResNet50 and ResNet101 have the same effective depth), which prevents rank-collapse of the solution space. Results in Figure 17 confirm this intuition. ResNets, unlike linear networks, *do not* exhibit a monotonic rank contracting behavior and the effective rank plateaus after 8 layers, regardless of using batch-normalization or not. Furthermore, preliminary experiments on least-squares using linear residual networks indicate that the effective rank of the solution space is also bounded by the number of layers in the shortest and longest path from the inputs to the outputs. A thorough study on the relationship between residual connections and rank is left for future work.

## J   Least-squares learning dynamics

The learning dynamics of a linear network change when over-parameterized. Here, we derive the effective update rule on least-squares using linear neural networks to provide motivation on why they have differing update dynamics. For a single-layer linear network parameterized by $W$, without bias, the update rule is:

$$W^{(t+1)} \leftarrow W^{(t)} - \eta \nabla_{W^{(t)}} \mathcal{L}(W^{(t)}, x, y) \tag{15}$$

$$= W^{(t)} - \eta \nabla_{W^{(t)}} \frac{1}{2}(y - W^{(t)}x)^2 \tag{16}$$

$$= W^{(t)} - \eta(W^{(t)}xx^T - yx^T) \tag{17}$$

Where $\eta$ is the learning rate. Similarly, the update rule for the two-layer network $y = W_e x = W_2 W_1 x$ can be written as:

$$W_1^{(t+1)} \leftarrow W_1^{(t)} - \eta(W_2^{(t)})^T(W_e^{(t)}xx^T - yx^T) \tag{18}$$

$$W_2^{(t+1)} \leftarrow W_2^{(t)} - \eta(W_e^{(t)}xx^T - yx^T)(W_1^{(t)})^T \tag{19}$$

$$\tag{20}$$

Using a short hand notation for $\nabla\mathcal{L}^{(t)} = W_e^{(t)}xx^T - yx^T$, we can compute the effective update rule for the two-layer network:

$$W_e^{(t+1)} = W_2^{(t+1)} W_1^{(t+1)} \tag{21}$$

$$= W_e^{(t)} - \eta \overbrace{(W_2^{(t)}W_2^{(t)T}\nabla\mathcal{L}^{(t)} + \nabla\mathcal{L}^{(t)}W_1^{(t)T}W_1^{(t)})}^{\text{first order } \mathcal{O}(\eta)} + \overbrace{\eta^2\nabla\mathcal{L}^{(t)}W_e^{(t)T}\nabla\mathcal{L}^{(t)}}^{\text{second order } \mathcal{O}(\eta^2)} \tag{22}$$

$$\approx W_e^{(t)} - \eta(P_2\nabla\mathcal{L}^{(t)} + \nabla\mathcal{L}^{(t)}P_1^T) \tag{23}$$

Where $P_i^{(t)} = W_i^{(t)}W_i^{(t)T}$ are the preconditioning matrices. The higher order terms can be ignored if the step-size is chosen sufficiently small.

**(General case)** For a linear network with $d$-layer expansion, the update for layer $1 \le i \le d$ is:

$$W_i^{(t+1)} \leftarrow W_i^{(t)} - \eta \overbrace{(W_d^{(t)}\cdots W_{i+1}^{(t)})^T}^{\text{weights} > i} \overbrace{(W_e^{(t)}xx^T - yx^T)}^{\text{original gradient}} \overbrace{(W_{i-1}^{(t)}\cdots W_1^{(t)})^T}^{\text{weights} < i} \tag{24}$$

Denoting $W_{j:i} = W_j \cdots W_{i+1} W_i$ for $j > i$, the effective update rule for the end-to-end matrix is:

$$W_e^{(t+1)} = \prod_{1 < i < d} W_i^{(t+1)} = \prod_{1 < i < d}(W_i - \eta W_{d:i+1}^{(t)T}\nabla\mathcal{L}^{(t)}W_{i-1:1}^T) \tag{25}$$

$$= W_e^{(t)} - \eta \sum_{1 < i < d} W_{d:i+1}W_{d:i+1}^T\nabla\mathcal{L}^{(t)}W_{i-1:1}^T W_{i-1:1} + \mathcal{O}(\eta^2) + \cdots + \mathcal{O}(\eta^d) \tag{26}$$

$$\approx W_e^{(t)} - \eta \sum_{1 < i < d} \underbrace{W_{d:i+1}W_{d:i+1}^T}_{\text{left precondition}} \underbrace{\nabla\mathcal{L}^{(t)}}_{\text{original gradient}} \underbrace{W_{i-1:1}^T W_{i-1:1}}_{\text{right precondition}} \tag{27}$$

The update rule for the general case has a much more complicated interaction of variables. For the edge $i = 1$ and $i = p$ the left and right preconditioning matrix is an identity matrix respectively.

## K   Rank-landscape

We visualize the effective rank landscape of the effective weights in Figure 18 and Gram matrices in Figure 19. We use single and two-layer linear networks for effective-rank landscape. We use two-layer, and four-layer ReLU networks for the Gram matrix and are constructed from 128 randomly sampled input data. For both methods, all the weights are sampled from the same distribution. The landscape is constructed by moving along random directions $u, v$. We observe that over-parameterized linear and non-linear models almost always exhibit a lower-rank landscape than their shallower counterparts.

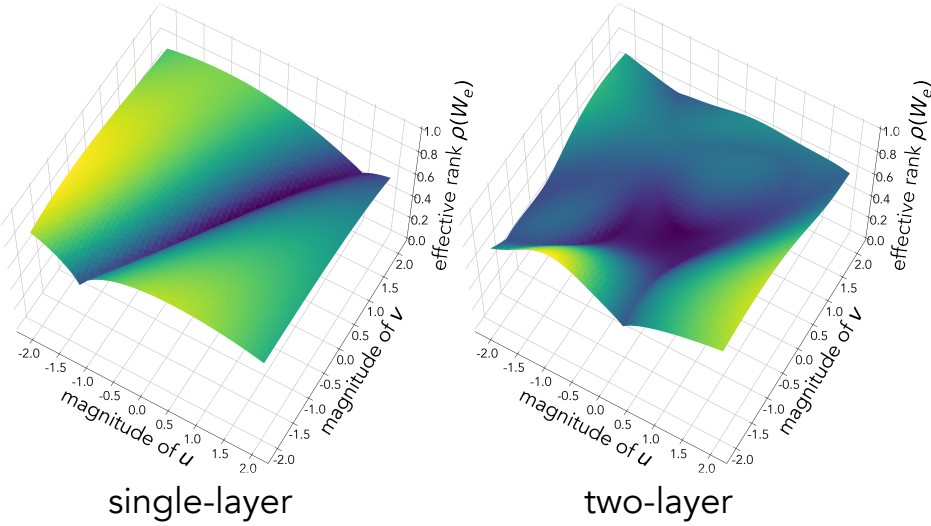

Figure 18: **Rank landscape**: The landscape of the effective rank $\rho$ of a linear function $W_e$ parameterized either by a single-layer network ($W_e = W$) or a two-layer linear network ($W_e = W_2 W_1$). The visualization illustrates a simplicity bias of depth, where the two-layer model has relatively more parameter volume mapping to lower rank $W_e$. Both models are initialized to the same end-to-end weights $W_e$ at the origin. Motivated by Goodfellow et al. (2015), the landscapes are generated using 2 random parameter directions $u, v$ to compute $f(\alpha, \beta) = \rho(W + \alpha \cdot u + \beta \cdot v)$ for the single-layer model and $f(\alpha, \beta) = \rho((W_2 + \alpha \cdot u_2 + \beta \cdot v_2) \cdot (W_1 + \alpha \cdot u_1 + \beta \cdot v_1))$ for the two-layer model ($u = [u_1, u_2]$, $v = [v_1, v_2]$).

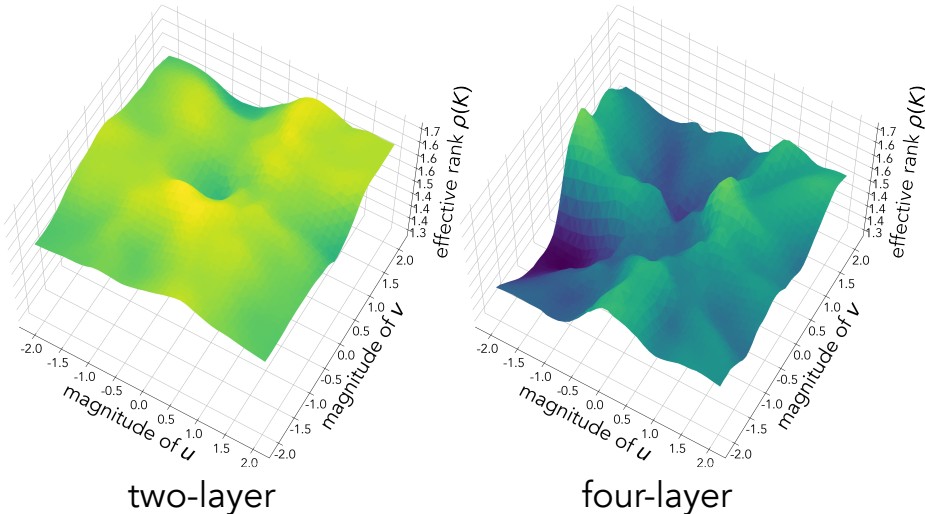

Figure 19: **Kernel rank landscape**: The landscape of the effective rank $\rho$ computed on the kernels constructed from random features.

## L    Relationship between weight and embeddings

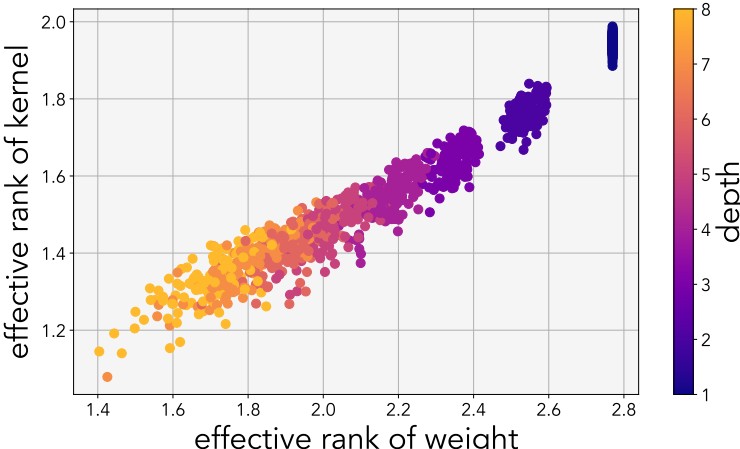

Figure 20: **Rank relation of kernel and weight**: Each point represents randomly drawn network. For each network, we compute the rank on the effective weight and also the linear kernel. The kernel is constructed from the `MNIST` dataset. The rank of the kernels and weights have a linear relationship.

We show that there is an almost one-to-one relationship between the effective rank of the weights and the effective rank of the Gram matrices in deep linear models. The figure plots this relationship for random deep linear networks applied to random subsets of the `MNIST` dataset. Moreover, it becomes apparent that the number of layers dictates the rank of the embedding as well as the weights.

## M  Noisy linear regression with least-squares

We extend our least-squares analysis under noisy observation to study the interaction between noise and the low-rank bias of the deep networks. We consider the standard linear regression with the least-squares objective with additive Gaussian noise:

$$Y = WX + \mathcal{N}(\mathbf{0}, \sigma \cdot \mathbf{I}) \tag{28}$$

In noisy linear regression, even if the intrinsic dimensionality of $W$ is low, the noise in the observation makes the relation between $X$ and $Y$ appear as full-rank. We consider two instantiations of noise injection, one in which the noise is sampled once (`static`) and another in which noise is resampled every iteration (`stochastic`). While the training loss is noisy, the test loss is computed on samples drawn from a noiseless system. In the presence of low-rank bias, even under noisy observations, deeper networks should be biased toward finding a low-rank solution. Hence, deeper networks should not overfit to the noise and result in better generalization.

To test this hypothesis, we repeat the least-squares experiment under noisy observation (see Figure 21). We set rank$(W) = 24$, for $W \in \mathbb{R}^{32 \times 32}$ and add varying levels of noise $\sigma \in \{0.1, 0.3, 0.5\}$ to the training labels. For all varying depths, we initialize the network to the same distribution and train the network with SGD and a learning-rate scheduler (decay-on-plateau). Note that the y-axis is scaled higher than the figures in the main paper to ensure we use the same y-axis across different noise levels. For all noise levels, we observed that deeper networks converge towards low-rank solutions, and we find that the sweet-spot depth that yields the best test performance varies for each setting. The experiments yielded a few surprising observations:

1. For static additive noise, the noise *can* be overfitted by the model. In this setting, we observed that shallower networks perfectly overfit to the noise while deeper networks cannot. Unlike the noiseless least squares, deeper networks resulted in better test performance. The shallower networks find solutions that are much higher effective rank. The observation implies that depth regularizes the model from overfitting to the noise.

2. For stochastic additive noise, the noise *cannot* be overfitted by the model. In this setting, we observed that the deeper networks found an even lower effective rank solution than the noiseless counterpart. Ultimately, while shallower networks perform worse than their noise-less counterpart, the deeper networks perform on par or better. We hypothesize that the stochasticity and simplicity bias leads to lower effective rank solutions.

In both settings of noisy least-squares, we observed that the simplicity bias of depth still persists. We observed that depth improves generalization performance by underfitting the noise in the data. This may explain why deep networks generalize well under weak supervision and corrupted labels. These observations further suggest that under noisy data, one should increase the depth to mitigate overfitting to noise.

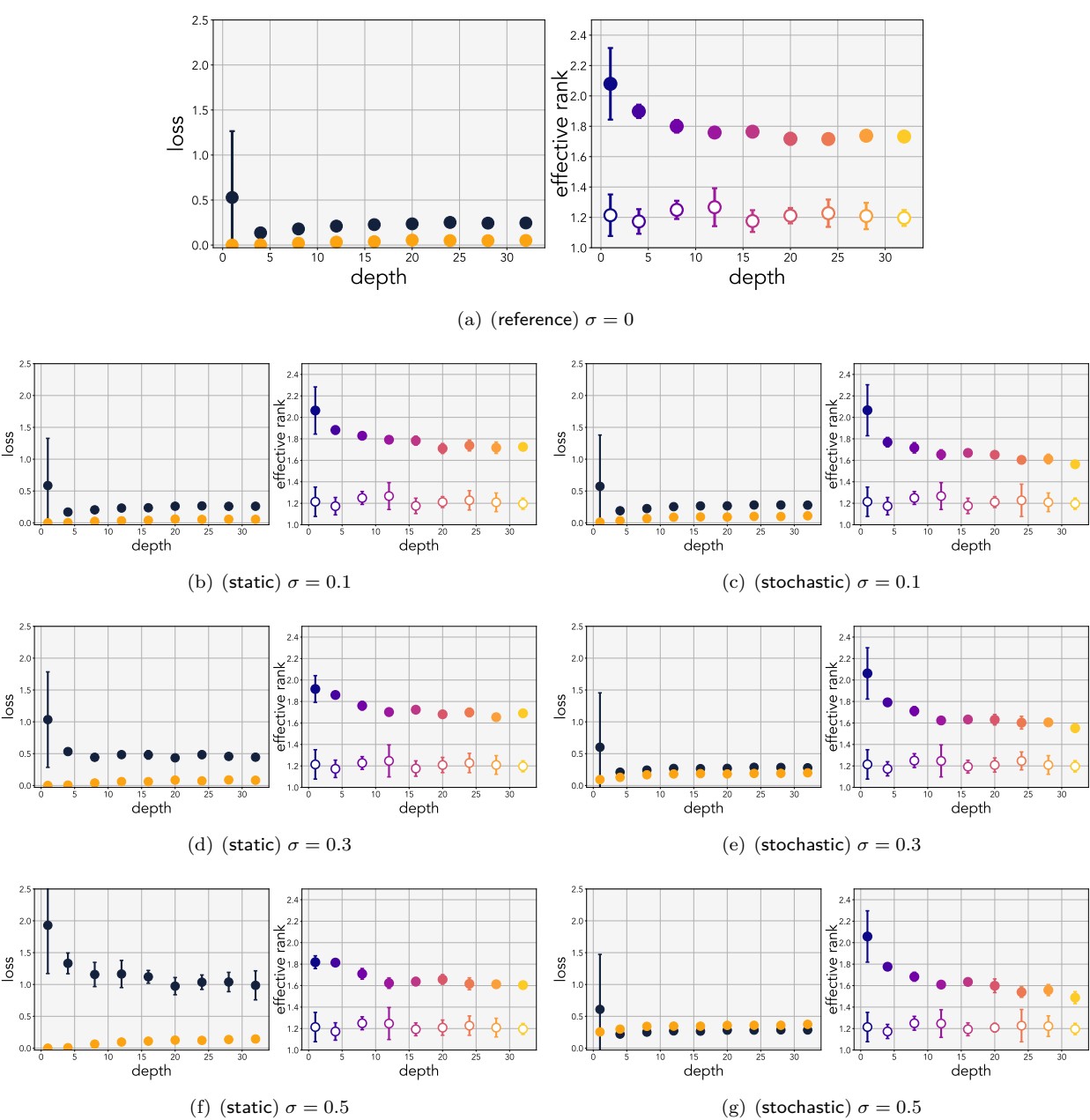

Figure 21: **Noisy least-squares:** Experiments investigating how noise affects the simplicity bias of depth.

