# OpenReview forum: "The Low-Rank Simplicity Bias in Deep Networks"
_TMLR — Accepted by TMLR_

### Review · Reviewer_6vzF · 2022-11-30

**Summary Of Contributions:**

The paper provides a thorough study on the bias towards low-rankness induced by increasing depth in neural networks. The main discoveries are

1). in linear regression, increasing the depth of linear layers leads to low effective rank embeddings;

2). in practical data sets, e.g., CIFAR10 and CIFAR100, augmenting linear layers can improve performance.

To support the above discoveries, many experimental results are provided as well as a related theoretical study.

**Audience:**

Yes

**Claims And Evidence:**

Yes

**Requested Changes:**

It is interesting to study noisy linear regression problems. Does the low-rank bias still hold when noise can potentially deteriorate the intrinsic structures?

The authors tried some small network architectures for CIFAR10 and CIFAR100 tasks. The performance gain of augmenting linear layers is statistically consistent. However, the improvement would be of more importance, if the improvement also appears on some state-of-the-art models.

In Figure 6, the training loss is not zero for deep networks. Thus the comparison of effective ranks might not be very convincing.

The statement of Theorem 3.1 is not very precise. The proof in Appendix H assumes all the matrices are square and then sends the dimension to infinity.

**Strengths And Weaknesses:**

=============== Strength ===============

The paper is well organized and relatively easy to follow, especially the main contributions and claims are highlighted and crystal clear.

Experiments are also dense and cover many aspects of the claims to provide comprehensive support.

In appendix, there is even a Q&A section to further define the scope of the paper and clarify statements.

In literature, implicit bias often associates with learning algorithms. This paper aims at a novel perspective and studies the implicit bias of network architectures. To my knowledge, this is an interesting and stimulating direction both to theorists and practitioners.

=============== Weakness ===============

Many of the empirical justifications are based on a simple noiseless linear regression model.

In real data experiments, the performance leaves a large gap to the state-of-the-art. For example, in CIFAR10, the best test accuracy in paper hits slightly less than 90%. However, ResNet and EfficientNet can easily achieve over 98% accuracy. Such a performance gap undermines the value of augmenting linear layers.

---

### Review · Reviewer_tssy · 2022-12-06

**Summary Of Contributions:**

In my opinion, the contributions of the paper can be summarised as follows
* The effective rank of representations of neural networks decreases as the network gets deeper. This holds both at initialisation and at convergence.
*  The paper argues this with a volume argument, which says that for deeper networks, the number of solutions that have a low rank are large.
* The paper also discusses its impact on generalisation.
* Finally, the paper proposes a regularisation objective using over-parameterisation to further exploit this bias.

**Audience:**

Yes

**Claims And Evidence:**

No

**Requested Changes:**

* Include the relevant citations and discuss what are the main differences to their work and which claims are novel to this paper and which are not etc.
* Table 1,2,3 needs error bars or some form of confidence intervals in order to be statistically significant.
* Table 2 needs a discussion of the various methods used to achieve the regularisations.  Please discuss how you regularise the various quantities and why the methods are sound. The method of Spectral Normalisation and Stable Rank Normalisation already exists in literature and they should be used to minimise stable rank (and if necessary spectral norm).
* There should also be a comparison of the over-parameterisation approach to reduce rank to the LR Layer used to reduce rank in Sanyal et. al. 2018.
* The baselines should have comparable parametric complexity as the number of parameters is often a confounder in DNN experiments.
* Finally, the authors should argue better why their experiments show the low rank bias at convergence holds in light of my arguemnts in the previous section.

**Strengths And Weaknesses:**

* I find Figure 1 to be a good demonstration of the difficulty of neural networks in fitting high rank ground truth functions. This is an interesting experimental control to observe this.

* The argument in Figure 2, that the effective rank of feedforward networks decrease with increasing depth is interesting and well reflected in Figure 2 for Linear and non-linear ReLU networks.


*However, this is only for randomly initialised networks which is not very surprising (or highly practically relevant) as the product of matrices or linear transformations are bound to decrease the rank unless all matrices are aligned which is a low probability event under random initialisation. What would be an interesting observation is if it also holds for neural network under convergence. This is what the authors target in Figure 3, but the results do not make that point as all it does it tries to get closer to the ground truth in its rank from its initialisation. Thus networks with smaller initialisation rank end up with larger initialisation rank and networks with larger initialisation rank end up with smaller ones. This is again expected of any function irrespective of what the true bias is. Given the observation regarding initialisation and that the ground truth is low-rank, Figure 3 is quite uninformative to me. If the initialisation is at rank x and the target is rank y and if the training gets you closer to the ground truth in terms of parameter recovery,  then is exactly what you would expect.

* This is further illustrated in Figure 6 but there once the depth is large enough to be relevant for deep network in practice (e.g. greater than 10), there is barely any change in rank.

1. Thus, my overall conclusion for this observation of rank vs depth is that the argument holds for initialisation, which immediately follows from standard linear algebra but the evidence provided in the paper for networks used in practice is not convincing.

2. My second and perhaps bigger criticism is that paper ignores many relevant works. It is already well known that neural networks are biased towards learning low dimensional representations. In particular Oyallon [2017] and Sanyal et. al 2018 study this specific problem. However, neither has been cited in this work. While Oyallon [2017]  studies how the  rank changes with depth, Sanyal et. al. 2018 also shows this phenomenon in practice, studies how a lower dimensionality improves robustness as well as generalisation and also proposes a specific regulariser to induce it in neural networks. All of these are claimed as novel in this work.

3. For linear networks, which is studied extensively in this paper has also been shown to tend towards low rank solutions in previous work with gradient flow. In particular, Ji and Telgarsky [2019 and 2020] showed that in linear networks with one dimensional output, gradient flow on commonly used exponential loss functions like cross entropy tend to networks where each layer is of rank one. Again both of these works are not cited.

4. There aren't enough details about what exactly is used for the algorithms in Table 1, 2, 3. Are all the networks of similar parametric complexity ? Or do the ones mentioned as "over-parameterised" actually have more parameters than the baseline? that would make the comparison very unfair.
Also what is the exact algorithm used to minimise the effective rank and the stable rank ? Are these from already existing algorithms like stable rank normalisation, if not, are there guarantees that they are actually minimising the stable rank etc ?


Oyallon Edouard. "Building a regular decision boundary with deep networks." 2017.

Sanyal, Amartya, et al. "Robustness via deep low-rank representations." 2018.

Ji, Ziwei, and Matus Telgarsky. "Gradient descent aligns the layers of deep linear networks." 2019

Ji, Ziwei, and Matus Telgarsky. "Directional convergence and alignment in deep learning."  2020

---

### Review · Reviewer_diZQ · 2022-12-07

**Summary Of Contributions:**

**Summary**

In this paper, the authors make a number of (empirical) observations about the inductive bias of deep neural network architectures. Ablating cross various choices of models, hyper-parameters, optimizers, datasets, and number of training epochs, they show that deeper networks generally lead to lower (effective) rank feature embeddings, a phenomenon that they call "low-rank simplicity bias".


**Audience:**

Yes

**Broader Impact Concerns:**

None to report.

**Claims And Evidence:**

Yes

**Requested Changes:**

**Critical**

- May be helpful to mathematically define (feature) embedding rank, at least informally, the first time it is introduced (Page 2). It only becomes clear in Section 2.3.
- I was mildly thrown off a few times by the somewhat fungible use of "low-rank" and "low-effective-rank". The authors make the case early on that the two are qualitatively equivalent, but that's not always the case. For example, Observation 3.1 (or Figure 2) is interesting in that low effective rank is exhibited at initialization but I would be surprised if random feature embeddings also exhibited low (standard) rank. Please reconsider carefully the use of "low-rank" in such places.

**Other comments**

- Consider remaking Figs 2,3,4,5 to be better accessible to color-challenged readers; it was a bit hard for me to separate the various curves. Different-shaped markers for each line plot might help.
- Page 8: Typo: "multilpications"
- Just a style point, but consider renaming 'FAQ' in the appendix as something else? Not common to see an FAQ list in papers.

**Strengths And Weaknesses:**

**Strengths**

* The authors study a problem of interest to the machine learning theory community.

* Similar observations about increased network depth inducing low-rank representations have been made in various contexts- for example, cf. several papers by Arora, Cohen, and other authors ca. 2019 onwards. Most of these earlier results are optimization-focused: they study a particular optimization method (usually, gradient descent) and show that the network weights obtained via this method eventually exhibit (some definition of) low-rankness. This paper can be viewed as an addition to the body of work, making the case that it is not just due to the optimization routine: the very representation itself inductively biases towards low-rankness. I found the case made by the paper to be fairly compelling.

**Weaknesses**

- My impression was that the paper made interesting observations, but did not provide any particular insights that were not known already. The NTK style results of Yang/Salman (2019), and earlier results by Valle-Perez et al (2018) already had made the case that deeper networks exhibit increasing spectral bias (although perhaps with a slightly different set assumptions than the current paper). Again, this may not be a dealbreaker for acceptance to TMLR: the empirics are fairly thorough and well thought out -- but wanted to highlight this nonetheless.

- I found that a lot of important details of the main paper were deferred to various Appendices, impeding the paper's flow. (For example, the main results of Section 2.3, 3.1, insights into the theory in 3.4, the stuff on linear overparameterization, etc are all key to the overall discussion.)

---

> ### Author Response · Authors · 2022-12-21
> **Response to Reviewer diZQ**
>
> **C1: May be helpful to mathematically define (feature) embedding rank in page 2**
> A: Thank you for the suggestion. We included text to define embedding rank on page 2.
>
> **C2: The interchangeable usage between “low-rank” and “low-effective-rank”**
> A: We apologize for any potential confusion. In light of the suggestion, we decided not to use these terms interchangeably and have changed the wording of the manuscript to reflect this.
>
> **C3: Minor comments: typos, naming suggestions, and figure color.**
> A: We have fixed our typos but have decided to keep the section name “Frequently asked questions”. Many of our peers, and even the reviewer 6vzF, have found it helpful as it does not overload the existing manuscript while providing a precise response to common concerns. For the coloring of the figures, we mainly used magma and virdis colormaps, which are perceptually uniform color map and has been known to be color-blind friendly (see https://cran.r-project.org/web/packages/viridis/vignettes/intro-to-viridis.html). As we do want our figures to be accessible to everyone, If the reviewer has a specific suggestion for the coloring, please let us know.
>
> **C4: Weakness: Similarity to Valle-Perez and Yang.**
> A: Our work is not the first to show the existence of the simplicity bias of deep networks, nor do we claim it to be. Our work is largely influenced by Valle-Perez et al., who observed that the complexity of the output is lower on deep randomly initialized boolean ReLU networks. The work of Yang et al. extends the observation beyond ReLU boolean networks by studying the simplicity bias in CK/NTK. Like Yang et al., our work builds on the existing theory. Hence we do not claim any new theoretical insights, but we do provide a connection in a potential connection RMT which no prior works have made. Our work and Yang et al. are similar in that the simplicity bias is due to the spectral bias of deep networks, but the contributions differ in many aspects. Yang et al. primarily provide theoretical insights on studying the simplicity bias of boolean networks with NTK. On the other hand, our work aimed to show that the simplicity bias also exists beyond these contrived settings by providing numerous empirical investigations on the low-rank simplicity bias in practical learning setups. To name a few, (1) we show the effect simplicity bias has on generalization and overfitting for least-squares and on linear and non-linear models, (2) we made a surprising observation that very deep networks cannot fit something as linear regression using standard optimization techniques (2) we empirically measure the PDF of the effective rank, and how it is affected by optimization, (3) provide a potential connection to the existing line of work in random matrix theory, (4) we show that this simplicity bias exists beyond the choice of optimizer and non-linearities, (5) we demonstrate one can leverage the bias to regularize the rank of the model via linear-overparameterization on classification tasks including ImageNet.

---

### Decision · Action_Editors · 2023-02-02

**Recommendation:** Accept with minor revision

**Comment:**

The key contribution of the paper is a thorough investigation of the low-rank bias of non-linear neural networks, with a focus on toy datasets. The Authors demonstrate that increasing depth in neural networks decreases a measure of the rank of model weights for both toy and realistic neural networks. Experiments suggest a causal relationship between the low-rank bias and generalization performance.

The paper received borderline reviews. Two reviewers voted for rejection. The third reviewer voted for acceptance after rebuttal.

One major criticism focused on the fact that results are not fully novel. My understanding is that similar observations can be found in the literature. For example, Figure 6 of Oyallon et al shows a metric related to effective rank decreasing with depth. Ji et al study linear networks (which are the significant focus of the paper) and show a form of low-rank bias under certain stringent assumptions. Somewhat related simplicity bias shows that deep neural networks have a bias toward learning simple functions (e.g. not learning high-frequency components of the input-output mapping).

The paper presents a potentially misleading narrative behind the key contributions. The Authors state in the introduction that “we stumbled upon a rather unexpected discovery that provides clues into why and when over-parameterization could help”. However, the discovery they have stumbled upon could be arguably deduced from existing results in the literature.

A view that is more in line with TMLR focus on correctness is presented in the Appendix. I agree that “our work is the first to extensively study the existence of low-rank bias in non-linear networks”. I would ask the Authors to revise the introduction to give a more objective view of the field.

My take is that the results are likely to be valuable to the community interested in the foundations of deep learning, even if the conclusions could be largely deduced from existing literature.

The second major criticism focused on the correctness of results. In particular, it indeed doesn’t seem to be surprising that in the toy task summarized in Figure 3 networks converge towards the rank of the training data. This is expected from the perspective of just fitting well the training set. According to my understanding, this indeed should be clarified in the paper or a different experiment should be run.

Another issue that was raised is that too much focus is on experiments on toy data. I think it is a valid criticism. However, image classification results (C10, C100, Imagenet) do suggest that there is a causal link between rank and generalization in deep neural networks, at least in image classification. (It was also raised that it is an unfair advantage that Authors added parameters. I would say it is OK because there is no nonlinearity between the added weight matrices).
Despite criticisms, I think there are important strengths to the paper. I mostly agree with Authors' statement “our work is the first to extensively study the existence of low-rank bias in non-linear networks”, with the caveat that results still focus very strongly on toy tasks.

All in all, I am recommending acceptance conditional on addressing some of the key criticisms. I would ask the Authors to rewrite  the introduction in order to make a clearer exposition of the existing works. I would also ask the Authors to add more context about Figure 3 or show analogous results on real data.

**Audience:**

The results should be interesting to the community interested in the foundations of deep learning. The paper presents a clean exposition to a specific form of implicit bias present in gradient-based training.

**Claims And Evidence:**

Most of the claims are generally supported by accurate, convincing and clear evidence. Certain issues were raised about Figure 3 and the introduction, see below.